# TDP-43 proteinopathy in ALS is triggered by loss of ASRGL1 and associated with HML-2 expression

Marta Garcia-Montojo [1], Saeed Fathi [1], Cyrus Rastegar[1], Elena Rita Simula [1,2], Tara Doucet-O'Hare [1], Y. H. Hank Cheng [1], Rachel P. M. Abrams [1], Nicholas Pasternack [1], Nasir Malik[3], Muzna Bachani[3], Brianna Disanza[1], Dragan Maric [4], Myoung-Hwa Lee[1], Herui Wang[5], Ulisses Santamaria[1], Wenxue Li[1], Kevon Sampson[1], Juan Ramiro Lorenzo [6], Ignacio E. Sanchez[7], Alexandre Mezghrani [1,8], Yan Li [9], Leonardo Antonio Sechi [2], Sebastian Pineda [10], Myriam Heiman [10], Manolis Kellis [10], Joseph Steiner[3] & Avindra Nath [1] ✉

TAR DNA-binding protein 43 (TDP-43) proteinopathy in brain cells is the hallmark of amyotrophic lateral sclerosis (ALS) but its cause remains elusive. Asparaginase-like-1 protein (ASRGL1) cleaves isoaspartates, which alter protein folding and susceptibility to proteolysis. *ASRGL1* gene harbors a copy of the human endogenous retrovirus *HML-2*, whose overexpression contributes to ALS pathogenesis. Here we show that ASRGL1 expression was diminished in ALS brain samples by RNA sequencing, immunohistochemistry, and western blotting. TDP-43 and ASRGL1 colocalized in neurons but, in the absence of ASRGL1, TDP-43 aggregated in the cytoplasm. TDP-43 was found to be prone to isoaspartate formation and a substrate for ASRGL1. ASRGL1 silencing triggered accumulation of misfolded, fragmented, phosphorylated and mislocalized TDP-43 in cultured neurons and motor cortex of female mice. Overexpression of ASRGL1 restored neuronal viability. Overexpression of HML-2 led to ASRGL1 silencing. Loss of ASRGL1 leading to TDP-43 aggregation may be a critical mechanism in ALS pathophysiology.

Amyotrophic lateral sclerosis (ALS) is the most prevalent adult-onset motor neuron disease with a median survival of 2–4 years[1]. Like other neurodegenerative diseases, ALS is considered a proteinopathy, which are characterized by abnormal protein folding and aggregation, leading to a loss of function and/or a gain of cytotoxic function of the proteins involved. The neuropathological hallmark of ALS is the proteinopathy of TAR DNA-binding protein 43 (TDP-43), an RNA/DNA-binding protein involved in RNA transcription[2], splicing, stability, transport and translation[3]. TDP-43 is predominantly located in the cell nucleus but in 97% of ALS patients it is aggregated in the cytoplasm of brain cells[4]. In those cytoplasmic inclusions TDP-43 is

misfolded[5], ubiquitinated[6] and phosphorylated[7]. Another feature of those inclusions is the presence of C-terminal fragments (CTFs) of TDP-43, mostly of 35 KDa (CTF35) and 25 KDa (CTF25)[8]. Seeded aggregation of TDP-43 in the cytoplasm may induce the mislocalization of the protein by impairing nuclear transport[9] and its deposition leads to RNA instability and may cause cell death by disrupting protein synthesis pathways[10]. ALS-linked mutations within the proteasome/autophagosome degradation pathways implicate failed TDP-43 clearance as a primary disease mechanism[11] but the molecular processes that lead to TDP-43 misfolding and impaired degradation remain elusive.

Formation of iso-aspartyl peptide bonds is a frequent source of non-enzymatic protein damage under physiological conditions[12]. Spontaneous deamidation of asparagine or dehydration of aspartate leads to the formation of isoaspartate residues, which can disrupt normal protein folding. Additionally, isoaspartates alter protease recognition of the peptide chains[13,14]. This disrupts neuronal proteostasis controlled by the ubiquitin-proteasome and autophagic-lysosomal pathways, both of which are impaired in ALS and other neurodegenerative diseases[11].

Asparaginase-like 1 protein (ASRGL1) is the only beta-aspartyl peptidase in mammals[15,16]. Its function is to degrade detrimental iso-aspartyl peptides. *ASRGL1* gene (chr11:62,337,448-62,393,412) is expressed in the brain and harbors an intronic copy in an antisense direction of the Human Endogenous Retrovirus-K (HERV-K), subtype HML-2, which has been associated with the pathogenesis of ALS[17–20]. Despite its enzymatic activity the role of ASRGL1 in neurological diseases is unknown.

Here we show that TDP-43 is a substrate of ASRGL1, whose expression is diminished in ALS brain samples. Loss of ASRGL1 triggers misfolding, fragmentation, phosphorylation and mislocalization of TDP-43 causing neuronal death. Moreover, we show that overexpression of HML-2 leads to ASRGL1 silencing and overexpression of ASRGL1 restores neuronal viability. Loss of ASRGL1 leading to TDP-43 proteinopathy may be a critical mechanism in ALS pathophysiology.

## Results

### RNA sequencing shows decreased expression of ASRGL1 and increased expression of TARDBP in ALS patients

Differential expression analysis was performed on RNA sequencing data from ALS patients ($n = 323$) and control ($n = 68$) motor cortex samples (Supplementary Table 1), provided by the ALS Consortium of the New York Genome Center. *ASRGL1* expression was significantly decreased in all patients compared to controls (estimated log2FC = −0.44, $p$ adj = 6.6E−9) (Fig. 1a). Expression of *TARDBP*, which encodes TDP-43, was significantly increased in ALS patients (estimated $log_2FC = 0.14$, $p$ adj = 0.0038) (Fig. 1a). Among the genes that were differentially expressed in ALS, we found several participating in the same biological processes associated with *ASRGL1*, according to the Gene Ontology (GO) Consortium (http://geneontology.org/), although their p-values were not as significant as for *ASRGL1* (Supplementary Fig. 1).

To clarify which brain cell types are mostly driving the differential expression of ASRGL1 in ALS and control individuals, we analyzed single-cell RNA sequencing data from a study by Dr Manolis and Dr Heiman's groups at the Massachusetts Institute of Technology (MIT)[21]. We compared the expression of *ASRGL1* and *TARDBP* between ALS individuals and controls (frontotemporal lobe dementia (FTLD) and non-neurological cases) in different subtypes of neurons and glia, in motor cortex (BA4) and in dorsolateral prefrontal cortex (BA9), as a control region. The frozen postmortem brain samples were obtained from the Mayo Clinic Neuropathology Laboratory (Jacksonville, FL). This included 73 age- and sex-matched individuals with sporadic ALS (sALS; $n = 17$), C9orf72-associated ALS (C9ALS; $n = 16$), sporadic FTLD (sFTLD; $n = 13$), C9orf72-associated FTLD (C9FTLD; $n = 11$) or neuropathologically normal individuals (PN; $n = 16$). All ALS and FTLD samples selected showed TDP-43 pathology. The results showed that *ASRGL1* expression was reduced below threshold of statistical significance in the motor cortex (BA4) of the sporadic ALS group in most subtypes of excitatory neurons from layer 2 to layer 6 (Supplementary Fig. 2). *TARDBP* was significantly elevated in the C9orf ALS group in 4 subtypes of excitatory neurons from layer 4 to layer 5 and from layer 5 to 6. It was also elevated but not statistically significant in the sporadic ALS group in several subtypes of excitatory neurons (Supplementary Fig. 2). In the sporadic and C9orf ALS groups, *ASRGL1* expression was also reduced mostly in astrocytes (GFAP−), although it did not reach statistical significance (Supplementary Fig. 3). *TARDBP*

showed statistically significant increased expression in the motor cortex of C9orf ALS patients in microglia, oligodendrocytes precursor cells (OPCs), and in the inhibitory neurons In5HT3aR VIP- and PV Basket. In the sALS group, *TARDBP* was elevated in several groups of neurons but without reaching statistical significance. It is important to note that single-cell data is poor at capturing relatively low-expressed genes, which both *ASRGL1* and *TARDBP* are in our data, so it is not surprising that the changes do not reach statistical significance.

*HERV-K* could not be analyzed in the bulk or single cell RNA sequencing analysis because most transposable elements are not captured by these assays, and those that are, are not sufficiently abundant to analyze. The assays only recover approximately 90–125 bases from the 3' end, which is not sufficient to accurately identify most transposable elements.

### Loss of ASRGL1 in the brain and in motor neurons derived from induced pluripotent stem cells (iPSCs) of patients with ALS and association with TDP-43 proteinopathy

To analyze the protein expression of ASRGL1 in ALS and controls, we stained formalin-fixed paraffin embedded (FFPE) brain cortical samples (Broadman area 6 (BA6); pre-motor cortex) from ALS ($n = 4$), Alzheimer's disease ($n = 4$), Multiple Sclerosis ($n = 4$), and non-neurological controls ($n = 4$) with an ASRGL1 antibody, using an automated staining system, to avoid differences due to manual staining. The expression of ASRGL1 was lower in ALS patients compared to the control groups (ANOVA; $p = 0.0004$) (Fig. 1b, e). To explore how the levels of ASRGL1 in affected areas compared to unaffected areas in ALS individuals, we stained slides of visual cortex (BA17) of the same individuals. ASRGL1 expression was slightly higher in BA17 than in BA6 (pre-motor cortex) but the difference did not reach statistical significance (Fig. 1c). To study a possible association between the loss of ASRGL1 in ALS and the presence of TDP-43 pathology, we stained contiguous slides of the same ALS brain samples stained for ASRGL1 with a TDP-43 antibody to measure the percentage of cells with cytoplasmic TDP-43. We found a negative correlation between the number of ASRGL1+ cells and the presence of cytoplasmic TDP-43 (Pearson $r = −0.84$; $p$ value = 0.008) (Fig. 1d). Images of TDP-43 staining are shown in Supplementary Fig. 4.

The levels of ASRGL1 were also greatly diminished in frozen brain samples (20 ALS and 20 normal controls, Supplementary Table 2), as analyzed by western blotting with two antibodies to ASRGL1: Atlas HPA029725 (Fig. 1f, g) and Atlas HPA055572 (Supplementary Fig. 5a, b). Differences were significant as well when samples were divided by sex: Control males vs. ALS males ($p = 0.0029$) and Control females vs. ALS females ($p = 0.01$) (two-way ANOVA, Tukey's post hoc test). By qPCR, the levels of *ASRGL1* RNA were also lower in the same frozen brain samples of ALS patients (Fig. 1h). When samples were analyzed by sex, statistical significance was maintained for males but not for females, probably because of the smaller sample size: Control males vs. ALS males ($p = 0.04$) and Control females vs. ALS females ($p = 0.2$) (two-way ANOVA, Tukey's post hoc test). Lower levels of ASRGL1 RNA were also found in fully differentiated motor neurons derived from iPSCs lines of ALS patients (Supplementary Table 3) (Fig. 1i) compared to motor neurons derived from control iPSC lines. These cells were mostly lower motor neurons (95%), as indicated by the expression of choline acetyltransferase (ChAT). (Characterization shown in Supplementary Fig. 6).

### Association of ASRGL1 loss and TDP-43 proteinopathy in brain samples of ALS patients

We selected 3 ALS FFPE brain samples (BA6 area) with the lowest levels of ASRGL1 as measured by immunostaining and 3 non-neurological controls to explore co-localization of ASRGL1 and TDP-43 by immunofluorescence. In control brains, ASRGL1 was easily detected and was co-localized with predominantly nuclear TDP-43 (Fig. 2a). In the cohort

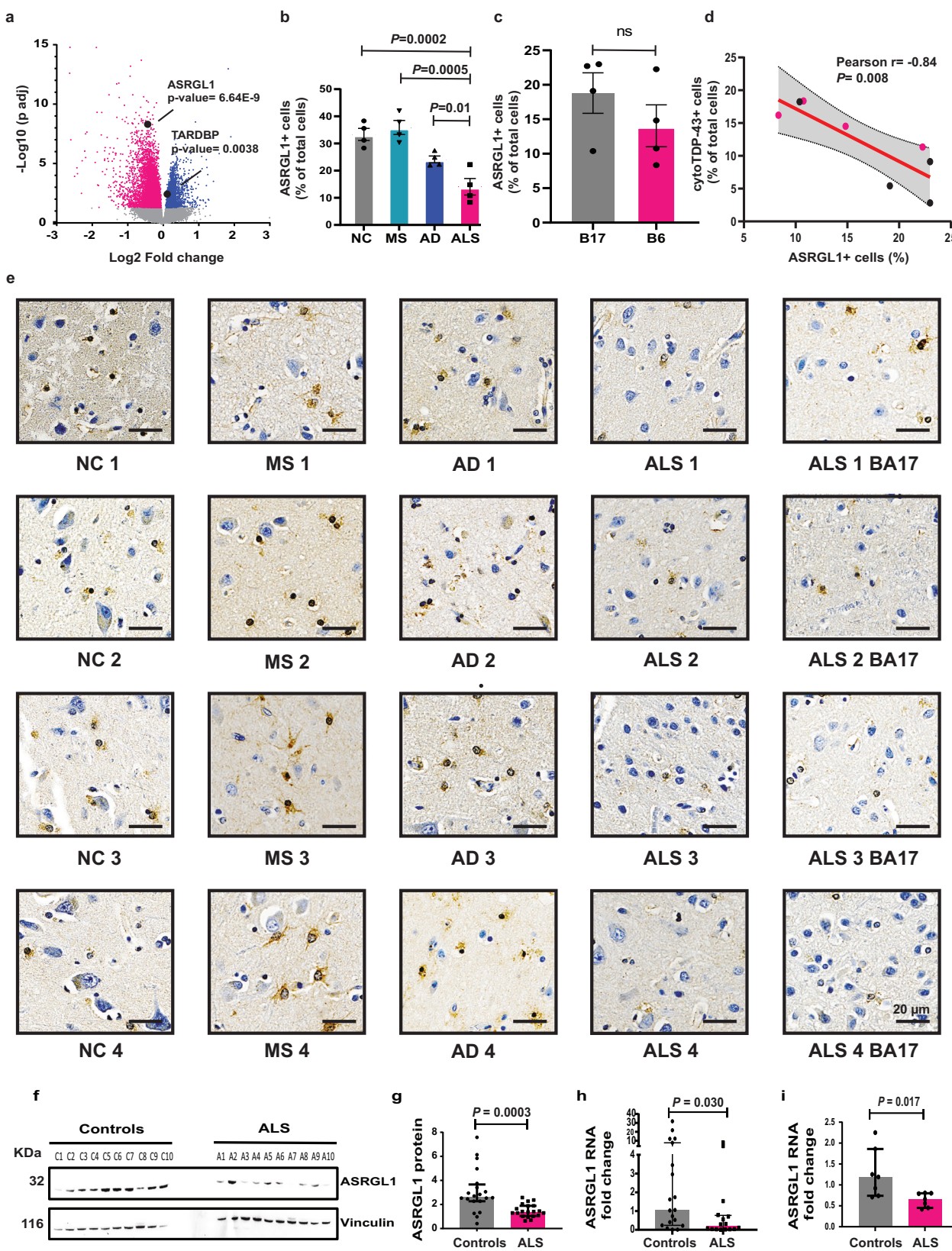

of ALS samples analyzed, pre-selected for low levels of ASRGL1, TDP-43 was found almost exclusively in the cytoplasm and, even though the levels of ASRGL1 were extremely low, the two proteins co-localized as well (Fig. 2a). The low expression of ASRGL1 was confirmed in these ALS samples by the number of neurons and astrocytes positive for ASRGL1, which was dramatically reduced in ALS brains compared to controls (Fig. 2b, d). The opposite was observed for TDP-43, with almost all neurons and astrocytes in this group of ALS patients presenting at least some cytoplasmic staining of TDP-43 (Fig. 2c, e). This reciprocal relationship was also evident at a cellular level, whereby neurons with cytoplasmic TDP-43 had low levels of ASRGL1 and cells with nuclear TDP-43, had perinuclear staining for ASRGL1 (Fig. 2f).

**Fig. 1 | Decreased levels of ASRGL1 in ALS patients associate with TDP-43 proteinopathy. a** Volcano plot showing the differential expression analysis (DEA) of RNA sequencing data from brain samples of ALS individuals ($n = 323$) and normal controls ($n = 68$). Blue dots: genes upregulated in ALS; red dots: genes downregulated in ALS; gray dots: genes with no statistically significant differences (Differential expression analysis (DEA)). Differentially expressed genes (DEGs) were counted as genes with a False discovery rate (FDR) adjusted $p$ value < 0.05. **b**, **c**, **e** ASRGL1 immunostaining of brain samples. Motor cortex samples (Broadman area 6) of ALS individuals ($n = 4$), multiple sclerosis individuals (MS) ($n = 4$), Alzheimer's disease (AD) ($n = 4$), and normal controls (NC) ($n = 4$) were immunostained for ASRGL1. Visual cortex samples (Broadman area 17) from the same ALS individuals were also stained for ASRGL1 and consecutive slides of those samples were stained for TDP-43. ASRGL1+ cells were quantified by automated microscopy and presence of cytoplasmic TDP-43 was analyzed manually in a blinded manner. **b** Percentage of cells positive for ASRGL1 in FFPE motor cortex samples (BA6), (one-way ANOVA (Mean ± SEM)). **c** Percentage of ASRGL1+ cells in BA6 and BA17 of the same ALS individuals (two-sided unpaired $T$-test; Mean ± SEM). **d** Correlation of the percentage of ASRGL1+ cells and the percentage of cells with cytoplasmic TDP-43 in ALS samples (red dots: BA6; black dots: BA17) (Pearson's $r$ test; Middle line represents best fit line and error lines represent 95% confidence bands of the best fit line). **e** Representative images of ASRGL1 immunostaining in brain. **f**, **g** Pre-motor cortex samples (BA 6) (20 ALS and 20 normal controls) were analyzed by western blotting to measure ASRGL1 protein (Atlas antibody 029725). **f** Representative image of a western blot for ASRGL1 and the loading control vinculin. **g** Levels of ASRGL1 protein (ratio ASRGL1/Vinculin) in ALS brains analyzed by Mann–Whitney test (Median (IQR)). **h** *ASRGL1* RNA levels were analyzed in brain samples (BA6) from ALS patients ($n = 19$) and normal controls ($n = 16$) by qPCR (Mann–Whitney test (Median (IQR)). **i** *ASRGL1* RNA levels were analyzed in differentiated motor neurons derived from iPSCs lines from ALS individuals ($n = 6$) and normal controls ($n = 6$), by qPCR (Mann–Whitney test (Median (IQR)). All pairwise comparisons in the Figure were performed with two-sided tests. Source data are provided as a Source Data file.

## Structural analysis of TDP-43 predicts isoaspartate formation through asparagine deamidation

To further study the association between ASRGL1 loss in ALS individuals and the presence of cytoplasmic TDP-43, we explored whether TDP-43 could be a substrate of ASRGL1 and if the loss of this enzyme could have a role in the formation of TDP-43 proteinopathy.

Isoaspartate formation occurs by asparagine deamidation or by aspartate isomerization in peptide chains most readily at sequences in which the side chain of the C-flanking amino acid of an asparagine or aspartate residue is relatively small and hydrophilic (Fig. 3a). Asn/Asp-Gly, Asn/Asp-Ser and Asn/Asp-His sequences, particularly when located in highly flexible protein regions, constitute "hot spots" for isoaspartate formation.

Analysis of the sequence of TDP-43 using the software NGOME-Lite, which predicts the deamidation of a protein under physiological-like conditions (pH close to 7, temperature = 310 K and moderate ionic strength), shows intrinsic sequences in TDP-43 with a propensity for deamination (Fig. 3b, top panel) and region-dependent structural protection (Fig. 3b, middle panel). It further predicts that 36.5% of full-length TDP-43 molecules undergo an asparagine deamidation reaction after 2 days (Fig. 3b, bottom panel). This is equivalent to a predicted deamidation half-life of 3 days for intact TDP-43, five-fold shorter than the average human protein[22].

Predicted "hot-spots" for asparagine deamidation are shown (Fig. 3b). The asparagine residue with the highest deamidation rate is N398, which is 16.5% deamidated by 2 days (average half-life of 7.8 days). Other residues prone to deamidation are N372, N265, N352 and N291. Analysis of the pathological C-terminal fraction, CTF35 of TDP-43 showed results similar to that of the full-length protein. The same deamidation hot spots were predicted in CTF25, with the addition of N301 due to the loss of structural protection in the shortest fragment. Interestingly, most of the asparagine residues prone to deamidate are present in the C-terminal region of TDP-43.

NGOME-Lite could not detect any rapid isomerization of aspartate residues to isoaspartate. This is because (1) Asp-Gly dipeptides isomerize slower than most Asn-Xaa dipeptides deamidate and (2) Asp-Gly dipeptides in TDP-43 are found in folded regions protected from isomerization reactions.

3D modeling of TDP-43 peptides with isoaspartate substitutions in the predicted hot spots showed structural differences from wild type TDP-43. Crystal structures of TDP-43 peptides were downloaded from the Protein Data Bank (PDB). The indicated asparagine residue was mutated to an isoaspartate, and the energy for both structures was calculated using Prime in the Schrödinger Suites Software package. In each case, isoaspartate formation twists the peptide chain, leading to a conformation with higher free energy as shown, for example, for Gly-Asn-Asn-Ser-Tyr-Ser to Gly-Asn-Iso(Asp)-Ser-Tyr-Ser (Fig. 3c). Thus, the isoaspartate modifications are predicted to reduce the thermodynamic stability of TDP-43 (Fig. 3c and Supplementary Data 1) (Structural Analysis of isoaspartate formation in TDP-43).

## Asparagine deamidation of TDP-43 in ALS brains

Frozen brain samples (Supplementary Table 2) (9 ALS; 9 normal controls) were analyzed by mass spectrometry after immunoprecipitation with a TDP-43 antibody (Supplementary Table 4) to identify deamidated asparagine residues in TDP-43, which were detected in 89% of ALS brains and in 55% of the controls.

## TDP-43 interacts with ASRGL1

ASRGL1 is a member of the N-terminal nucleophile (Ntn) family that hydrolyzes L-asparagine and isoaspartyl-peptides. The protein folds into an αβ–βα sandwich homodimer that auto-cleaves at the G167–T168 bond, resulting in the active form of the enzyme. T219 is the active site for substrate hydrolysis[23]. To determine whether ASRGL1 is involved in the clearance of detrimental isoaspartates of TDP-43 we first analyzed their interaction "in silico" and found that TDP-43 could dock into the ASRGL1 active site (Supplementary Fig. 7). The interaction of both proteins was further studied by a proximity ligation assay (PLA) (Fig. 3d) in HEK-293 cells co-transfected with ASRGL1 and TDP-43 plasmids. When antibodies against both proteins were used together an intense fluorescence was emitted, indicating the interaction, mainly in the perinuclear region (Fig. 3e and Supplementary Movie 1). This could not be seen in any of the controls (no primary antibodies, only TDP-43 antibody, only ASRGL1 antibody or TDP-43 antibody + an isotype IgG) (Supplementary Table 4 and Fig. 3e, f).

To measure the binding affinity of ASRGL1 and TDP-43 we used differential scanning fluorimetry (DSF) known as protein thermal shift assay. At pH 7.4, the assay showed reproducible protein melting temperatures (Tm). Upon incubation of recombinant ASRGL1 with increasing concentrations of recombinant TDP-43, there was a concentration-dependent shift in Tm of about 10 °C with 1 μM TDP-43, and an apparent binding affinity (EC50) of 36.1 nM (Fig. 3g). In contrast, no binding was seen with chaotropic denatured TDP-43.

To determine whether TDP-43 and ASRGL1 interact in the brain and to further confirm the lower levels of ASRGL1 in the brain of ALS individuals we performed a PLA assay on FFPE cortex samples from ALS individuals and controls. The level of PLA fluorescence was significantly higher in controls than in ALS patients (Fig. 3h, i), indicating higher levels of interaction between the proteins, likely due to the higher level of ASRGL1. In controls, the PLA signal presents in a ring around the nucleus, suggesting that the interaction between the two proteins takes place in this region (Fig. 3h). We also attempted to confirm the interaction of the two proteins in the brain by co-immunoprecipitation. However, the results were inconsistent, likely

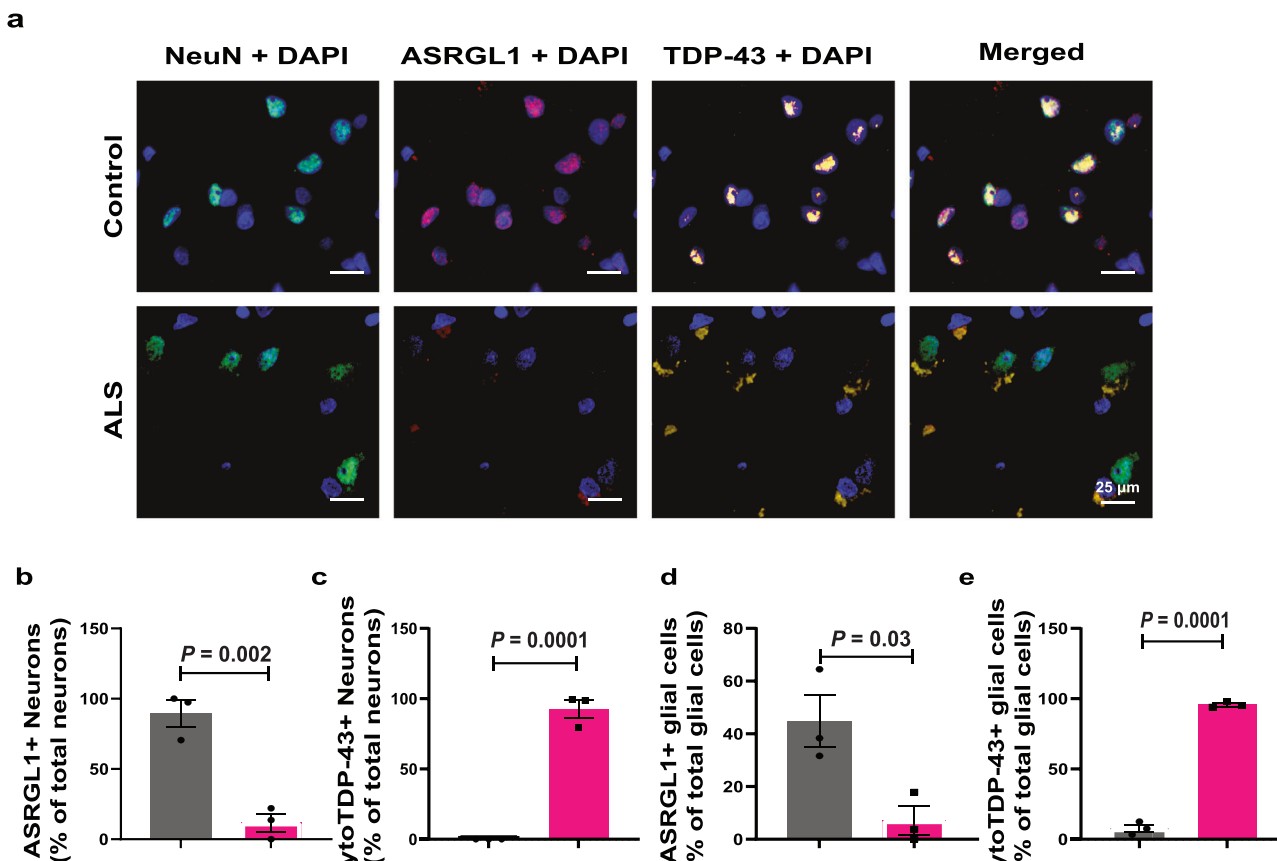

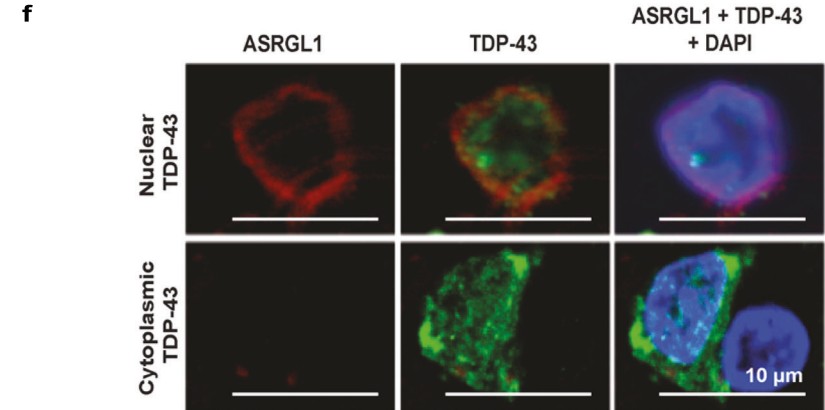

**Fig. 2 | Association of ASRGL1 loss and TDP-43 proteinopathy in brain samples of ALS patients. a**–**f** FFPE brain cortex samples from 3 pre-selected ALS individuals with very low levels of ASRGL1 and 3 normal controls were stained by immunofluorescence with ASRGL1 and TDP-43 antibodies. **a** Representative immunofluorescence images of brain cortex showing ASRGL1 and TDP-43 in an ALS individual and a normal control. **b** Percentage of neurons positive for ASRGL1 in normal controls and ALS brain samples. **c** Percentage of neurons showing cytoplasmic TDP-43. **d** Percentage of glial cells positive for ASRGL1 in controls and ALS patients. **e** Percentage of glial cells showing cytoplasmic TDP-43. Data in (**b**–**e**) analyzed by two-sided Unpaired $T$-test and represented as mean ± SEM. **f** Amplified image of brain cells of the same cortical brain sample of an ALS patient showing nuclear TDP-43 staining associated with perinuclear staining of ASRGL1 (upper panels) and loss of ASRGL1 associated with cytoplasmic TDP-43 (lower panels). Source data are provided as a Source Data file.

because ASRGL1 is a protease, and the interactions may be too transient to be detected by this method.

## ASRGL1 silencing results in neurotoxicity

To determine whether the inhibition of ASRGL1 affects neuronal viability, human neurons derived from induced pluripotent stem cells (iPSCs) modified to stably express tdTomato protein were transfected with a combination of four different shRNAs against ASRGL1 and monitored for cell viability by counting the fluorescent cells by automated fluorescent microscopy. All combinations of shRNAs increased neuronal cell death compared to scrambled shRNAs (Fig. 4a, b). A similar effect was seen when mouse neurons stably expressing GFP

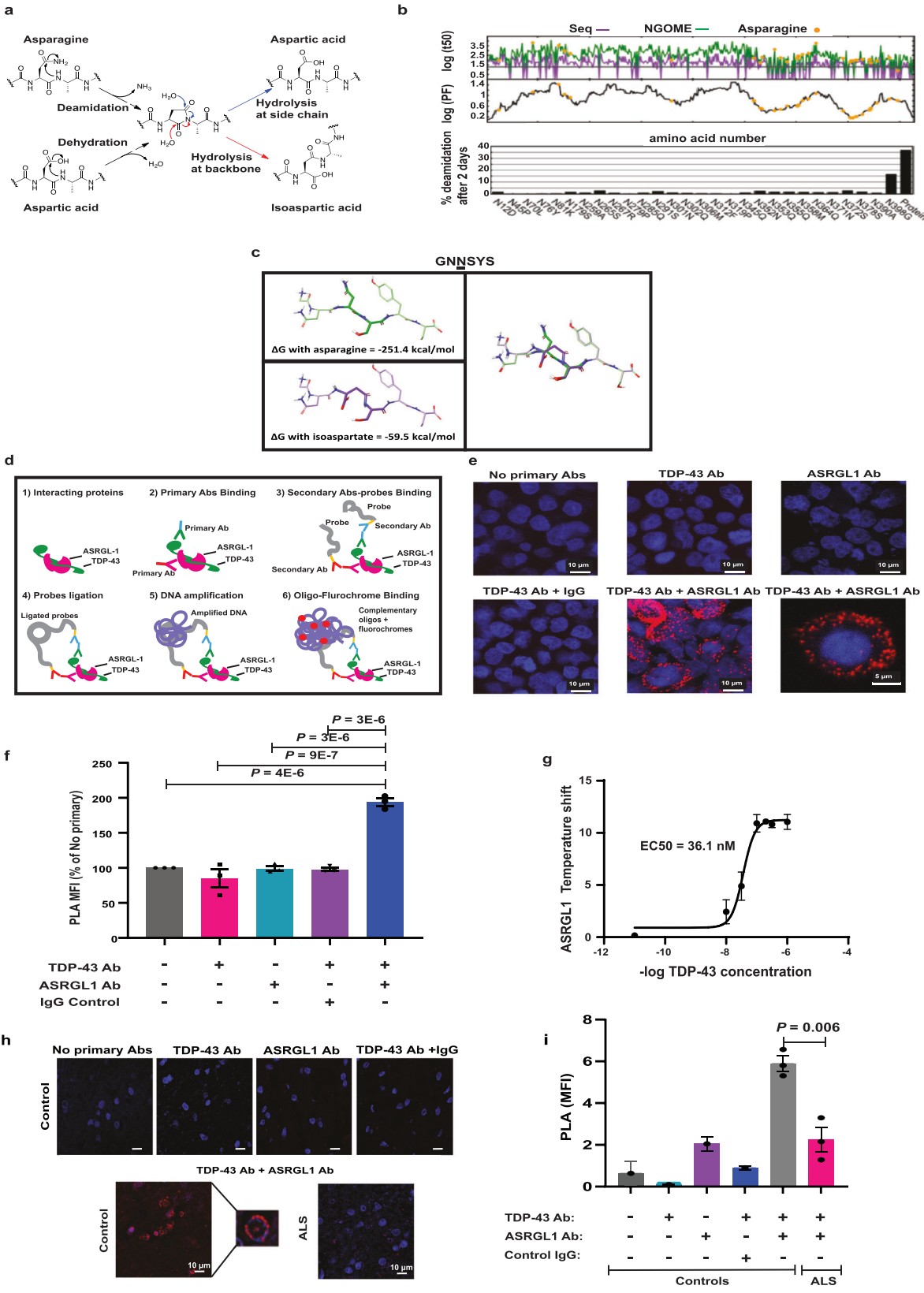

were transfected with different combinations of four shRNAs against the murine sequence of ASRGL1 (Fig. 4c, d). Single shRNAs also significantly decreased neuronal viability, although to a lesser extent than the combinations (Supplementary Fig. 8a, b). We confirmed these results in another set of experiments where cell viability was measured by fluorescence spectrometry. Here cells were exposed to a resazurin-

based solution which is reduced to resorufin inside the cell and emits a red fluorescence and thus serves as an indicator of live cells (Alamar-blue, Thermo Fisher) (Supplementary Fig. 9). For further experiments involving silencing of ASRGL1, we used the combination of shRNAs B, C, and D. Validation of ASRGL1 silencing was performed by PCR and western-blotting (Supplementary Fig. 10).

**Fig. 3 | TDP-43 is predicted to form isoaspartates through asparagine deamidation and TDP-43 interacts with ASRGL1. a** Spontaneous formation of isoaspartate by asparagine deamidation or aspartate isomerization. **b** Prediction of TDP-43 asparagine deamidation under physiological conditions using the NGOME algorithm[81]. Top panel: Predicted deamidation half-lives (time point at which the residue is deaminated in 50%) considering sequence propensities (purple line) and sequence propensities + structural protection (green line). Orange circles highlight actual asparagine residues; the lines are the predictions for all positions of the query sequence[81]. Middle panel: Structure protection factor (PF) as a function of residue number[81]. Bottom panel: Percentage of deamidation predicted for asparagine residues after 2 days. **c** Example of 3D modeling of a TDP-43 peptide with isoaspartate substitution in a "hot spot" predicted by NGOME-Lite, showing the variation in the Gibbs free energy values, indicative of thermodynamic instability. **d** Diagram explaining Proximity ligation assay (PLA). **e, f** PLA was performed in HEK 293 cells transfected with ASRGL1 and TDP-43 plasmids and incubated with antibodies: no primary, only TDP-43, only ASRGL1, TDP-43 + IgG and ASRGL1 + TDP-43.

**e** Representative PLA images of HEK 293 cells transfected with ASRGL1 and TDP-43 plasmids and incubated with ASRGL1 + TDP-43 antibodies and controls. **f** Comparison of percentage of PLA mean fluorescence intensity (PLA MFI) in relation with "No primary" control. (Brown-Forsythe ANOVA test/Bonferroni correction; Mean ± SEM; number of experimental replicates = 3). **g** Interaction of ASRGL-1 with TDP43 (recombinant proteins) using thermal shift assays. Tm value was calculated from the derivative of the thermal melt curves (Sigmoidal, Four-parameter logistic curve (4PL); Mean (95% CI); number of experimental replicates = 3). **h, i** PLA was performed in brain samples of ALS individuals (n = 3) and normal controls (n = 3) with TDP-43 and ASRGL1 antibodies. Negative controls were performed on a normal control brain since the expression of ASRGL1 is higher in normal controls. **h** Representative images of PLA in a normal control and an ALS patient brain stained with TDP-43 + ASRGL1 antibodies and the negative controls. **i** PLA MFI comparison (one-way ANOVA with Bonferroni correction (Mean ± SEM). All pairwise comparisons in the Figure were performed with two-sided tests. Source data are provided as a Source Data file.

Mouse neurons were also transduced with AAV9 viral particles packing a vector encoding four shRNAs (Supplementary Fig. 11a), which resulted in ASRGL1 knock-down as demonstrated by immunoblotting (Supplementary Fig. 11b) and caused neurotoxicity in a dose-responsive manner (Supplementary Fig. 11c).

## ASRGL1 silencing causes TDP-43 proteinopathy in human neuronal cultures

A loss of ASRGL1 function would lead to the accumulation of isoaspartates in its substrate proteins, which could contribute to TDP-43 proteinopathy by increasing protein misfolding and by impairing protein degradation[13]. To measure misfolded TDP-43 we used an intrabody against a molecular epitope (D247) not exposed in the physiological form[24] (Supplementary Fig. 12a, b). The intrabody was validated using ALS-associated TDP-43 mutants prone to misfolding (Supplementary Fig. 12c, d). In previous studies, this intrabody has been shown to recognize mislocalized or misfolded TDP-43 in spinal cord sections from ALS individuals and cultured cells, but not wild-type TDP-43 in the nucleus[25]. We co-transfected iPSC-derived neuronal cultures with plasmids encoding an intrabody against misfolded TDP-43, full-length TDP-43, and ASRGL1 shRNAs or scrambled shRNAs, and performed co-immunoprecipitation of the intrabody-TDP-43 complexes followed by western blot analysis. ASRGL1 silencing with shRNA led to accumulation of misfolded TDP-43 in neuronal cultures (Fig. 4e, f). As a control, we treated cells with a proteosome inhibitor, MG132 and found accumulation of misfolded TDP43 which was recognized by the intrabody (Fig. 4e, f).

TDP-43 cytoplasmic inclusions in ALS are characterized by the presence of full length TDP-43 together with CTFs of the protein, mainly of 25 and 35 kDa[4]. These pathological fragments are prone to aggregation and are degraded by proteases in the proteasome and autophagosome pathways[26]. To determine how ASRGL1 silencing affects the turnover of TDP-43 CTFs, we transfected iPSC-derived human neurons with ASRGL1 shRNAs or scrambled shRNAs and with a plasmid encoding TDP-43. TDP-43 CTFs accumulated in the ASRGL1-silenced cells as analyzed by western-blots (Fig. 4g, h). Neurons treated with MG132 were used as a positive control. Another characteristic of TDP-43 inclusions is hyperphosphorylation of the protein, although its role in neurodegeneration is disputed: some authors suggest that it may disrupt signaling pathways or affect the trafficking of TDP-43[27], while others indicate that hyperphosphorylation might be a protective cellular response to counteract TDP-43 aggregation[28]. ASRGL1 silencing increased the accumulation of phosphorylated TDP-43 in human neuronal cultures (Fig. 4i, j) when neurons were co-transfected with the shRNAs and with a plasmid encoding TDP-43. It is important to note that for the measurement of misfolded TDP-43 and TDP-43 fragments the blots had to be overexposed, due to the low levels of those pathological

proteins. Moreover, an increase in cytoplasmic localization of TDP-43 was seen in ASRGL1-silenced neurons compared to those transfected with scrambled shRNA (Fig. 4k, l). In neurons where ASRGL1 was visible, TDP-43 showed a nuclear localization, while in neurons with no visible or reduced ASRGL1, TDP-43 was cytoplasmic (Fig. 4l).

To determine whether ASRGL1 knockdown affects the function of TDP-43, we studied a gene whose splicing is controlled by TDP-43. TDP-43 controls a cryptic exon-splicing event in UNC13A. In ALS, the loss of TDP-43 from the nucleus results in the inclusion of a cryptic exon in UNC13A mRNA and reduced UNC13A protein expression[29]. We transfected iPSC-derived neuronal cultures with ASRGL1 shRNAs or with scrambled shRNAs and measured the cryptic exon of UNC13A by PCR and UNC13A protein by western-blotting. The results show that ASRGL1 knockdown causes an increase in the levels of the cryptic exon (Fig. 4m) and a consequent decrease in the protein levels (Fig. 4n, o), indicating a malfunction in its splicing, which is driven by TDP-43.

## Protein degradation is impaired in ASRGL1-silenced neurons

The presence of isoaspartates strongly impairs the degradation of beta-amyloid peptides by proteases[13]. We hypothesized that accumulation of isoaspartates due to ASRGL1 loss could affect neuronal proteostasis. Accumulation of ubiquitinated proteins was seen in ASRGL1-silenced neurons, indicating impairment of degradation by the proteasome system (Fig. 5a, b). To further determine how ASRGL1 silencing affects the degradation of TDP-43, we treated iPSC-derived neuronal cultures with cycloheximide (CHX) to block protein synthesis and analyzed TDP-43 levels by western-blot before and after 6 h of CHX treatment. In this case the neurons were not transfected with TDP-43 to focus only on the degradation of the endogenous protein. TDP-43 levels did not change significantly in the neurons treated with scrambled shRNA, while in ASRGL1-silenced neurons TDP-43 accumulated, indicating impaired degradation (Fig. 5c, d). Although the effect of ASRGL1 knockdown on the accumulation of TDP-43 was highly significant compared to the cells transfected with scrambled shRNA, it is difficult to compare it with the effect of MG132. MG132 was applied 3 h prior to adding CHX, so an accumulation of TDP-43 was already seen at the time of adding CHX. An increase in the loading control, vinculin, could also be seen (Fig. 5c). Hence, change in TDP-43 in the MG132-treated cells could not be accurately calculated.

The endoplasmic reticulum (ER) plays an important role in the maintenance of proteostasis through the unfolded protein response (UPR), which is activated in most neurodegenerative disorders[30]. The UPR is activated in response to an accumulation of misfolded proteins in the lumen of the ER (ER stress). Splicing of the X-box binding protein (sXBP1) is a marker of UPR activation[30]. An increase in sXBP1 was detected in ASRGL1-silenced neurons, indicating UPR activation and thus, accumulation of misfolded proteins (Fig. 5e, f).

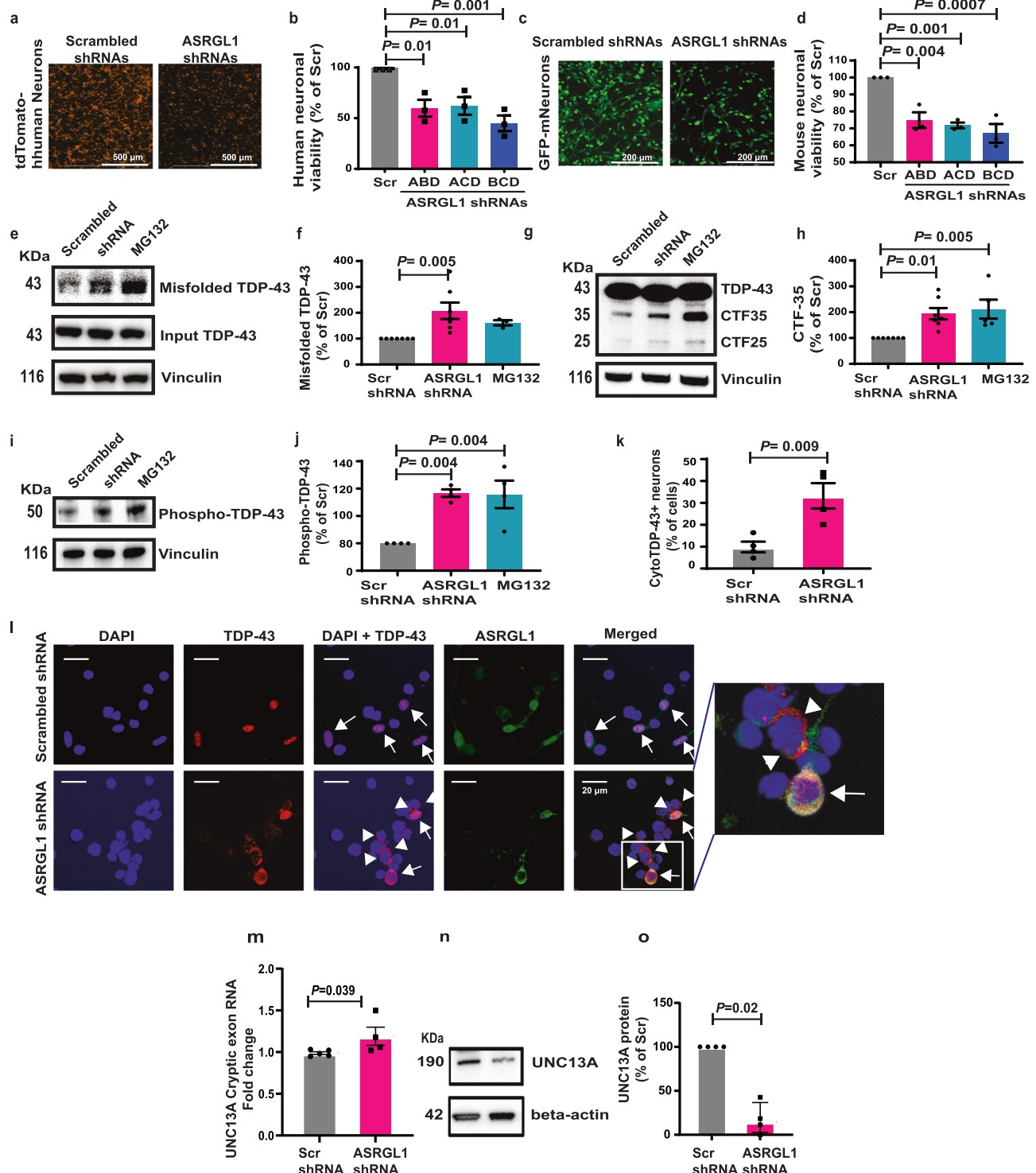

## ASRGL1 rescues TDP-43-proteinopathy and associated neurotoxicity

We transfected iPSC-derived neuronal cultures with a TDP-43 plasmid and with shRNAs against ASRGL1 and co-transfected them with increasing concentrations of an ASRGL1-encoding plasmid or with pcDNA as a control. Then, we analyzed the number of cells with cytoplasmic TDP-43 by immunofluorescence confocal microscopy. Around 15% of neurons knocked down for ASRGL1 showed cytoplasmic TDP-43, while cells transfected with increasing concentrations of the ASRGL1 plasmid, showed significantly lower percentage of cells with the cytoplasmic aggregates (Fig. 6a). Recovery of cell viability as

measured by applying Alamarblue followed by fluor spectrometry could also be seen in the neurons transfected with the two highest concentrations of ASRGL1 plasmid compared to the control cells (Fig. 6b).

To determine if TDP-43 with isoaspartyl modifications were neurotoxic, we transfected tdTomato-neuronal cultures with wild type and isoaspartate-containing C-terminal TDP-43 peptides (only peptides but not the full sequence of the protein could be synthetized with this post-translational modifications due to technical limitations) (Creative Peptides (NJ, USA) (TDP-43 wild-type: SYSGSNSGAAIGWGSAS-NAGSGSGFNGGFGSSMDSKSSGWGM; Iso-aspartyl-modified: [β-Asp]

**Fig. 4 | ASRGL1 silencing causes TDP-43 proteinopathy and neuronal death.**
**a, b** IPSc-derived human neurons (normal donor) line stably expressing tdTomato protein and (**c, d**) Mouse neurons derived from a commercial neural stem cell line stably expressing GFP were transfected with ASRGL1 shRNAs or scrambled shRNAs. Fluorescent neurons were counted automatically after 48 h. **a, c** Representative fluorescence microscopy images. **b, d** Comparison of cellular viability (Percentage of counted cells/field in relation to scrambled shRNAs) (Brown-Forsythe ANOVA test with Bonferroni correction; Mean ± SEM; number of experimental replicates = 3). **e, f** Misfolded TDP-43 was analyzed in IPSc derived (normal donor) neuronal cultures co-transfected with an intrabody against misfolded TDP-43, full-length TDP-43, and ASRGL1 shRNAs or scrambled shRNAs, by co-immunoprecipitation and western blotting. Proteasome inhibitor MG132 was used as positive control. **e** Image of a TDP-43 blot. Due to the low levels of misfolded protein, blots were exposed longer. **f** Comparison of misfolded TDP-43 levels (Brown-Forsythe ANOVA test with Bonferroni correction; Mean ± SEM; number of experimental replicates = 7). **g, h** Neuronal cultures were transfected with TDP-43 and ASRGL1 shRNAs or scrambled shRNAs and analyzed by western blotting. **g** Image of a TDP-43 C-terminal fragments (CFT) blot. Due to the low levels of CFTs, blots were exposed longer. **h** Comparison of CTF levels (Brown-Forsythe ANOVA test with Bonferroni correction; Mean ± SEM; number of experimental replicates = 7). **i–o** Neuronal cultures were transfected with ASRGL1 shRNAs or scrambled shRNAs and analyzed by western blotting. **i** Image of a phosphorylated TDP-43 blot. **j** Comparison of phosphorylated TDP-43 levels (Brown-Forsythe ANOVA test with Bonferroni correction; Mean ± SEM; number of experimental replicates = 4). **k** Comparison of the percentage of neurons with cytoplasmic TDP-43 (two-sided unpaired T-test; Mean ± SEM. **l** Confocal microscope images of nuclear TDP-43 (arrows) and TDP-43 cytoplasmic inclusions (arrow heads) in neurons. **m** RNA levels of UNC13A cryptic exon, by qPCR. **n** Image of a UNC13A protein blot. **o** Comparison of UNC13A protein levels in percentage relative to scrambled shRNA transfected cells (Mann–Whitney test; Mean ± SEM; number of experimental replicates = 4). All pairwise comparisons in the Figure were performed with two-sided tests. Source data are provided as a Source Data file.

SYSGSNSGAAIGWGSASNAGSGSGFNGGFGSSMDSKSSGWGM). Peptides instead of plasmids were used since the isoaspartate is a post-translational modification that cannot be induced by plasmid transfection. Both peptides triggered neurotoxicity, as measured by cell count and neurite length, which was prevented by transfection with a vector encoding ASRGL1 (Fig. 6c, d). The protective effect of ASRGL1 expression was higher against the isoaspartyl peptides although it also protected against wild-type peptides, possibly because it prevented formation of endogenous isoaspartates (Fig. 6d).

### ASRGL1 silencing triggers death of motor neurons and TDP-43 proteinopathy in mice

To analyze the effect of ASRGL1 silencing in vivo, ten 3-month-old adult C57BL/6 mice received a stereotaxic injection of AAV9 viral particles packing ASRGL1 shRNAs or scrambled shRNAs (Fig. 7a). Brain sections were immunostained to confirm ASRGL1 silencing (Supplementary Fig. 13). The number of neurons (Fig. 7b, c) and motor neurons (Fig. 7b, d) were significantly lower in the mice treated with ASRGL1 shRNAs, consistent with a neurotoxic effect of ASRGL1 depletion. Moreover, the number of cells showing cytoplasmic mislocalization or a complete loss of nuclear TDP-43 was markedly increased in the ASRGL1-silenced brains (Fig. 7e, f).

### HERV-K (HML-2) overexpression decreases ASRGL1 levels

HERV-K (HML-2) is a human endogenous retrovirus which has been associated with ALS pathogenesis[18–20,31–33]. It has one copy inserted in an *ASRGL1* intron in opposite direction, between the 4th and 5th exons (chr11:62368491-62383091; GRCh38/hg38) (Fig. 8a). Higher levels of HML-2 have been detected in ALS brains[17,18] and its overexpression causes neuronal death; although HML-2 activation might not be exclusive to ALS[34]. The *HML-2* copy is in the opposite strand of the ASRGL1 gene thus, the RNA of *HML-2* is complementary to the RNA of *ASRGL1*. If *HML-2* is activated, its reverse transcriptase could form a cDNA from the *HML-2* transcript[35] which could also potentially bind to the *ASRGL1* RNA. We thus hypothesized that HML-2 activation may downregulate ASRGL1 and may contribute to its depletion in ALS. Interestingly, *ASRGL1* is the only gene with an intronic insertion of *HML-2* whose expression has shown a correlation with the expression of its overlapping provirus[36], possibly indicating read-through transcription due to intron retention. Thus, we evaluated whether the overexpression of HML-2 could affect the expression of ASRGL1.

Transfection of iPSC-derived neuronal cultures with increasing concentrations of a plasmid encoding a consensus sequence of HML-2 caused a decrease in *ASRGL1* RNA in a dose-responsive manner as determined by qPCR (Fig. 8b). ASRGL1 protein also decreased in HML-2-transfected neurons (Fig. 8c, d). To determine how endogenous HML-2 expression affects ASRGL1 expression, we transfected iPSC-derived neuronal cultures with a plasmid encoding CRISPR/dCAS9 (VP160) and guide RNAs to the LTR of HML-2. When endogenous *HML-2* loci were activated, the levels of ASRGL1 transcript (Fig. 8e) and protein (Fig. 8f) decreased. Treatment of neural stem cells with the reverse transcriptase inhibitor zidovudine (AZT) increased the levels of ASRGL1 protein in a dose-responsive manner (Fig. 8g, h).

### ASRGL1 is neuroprotective against HML-2

Our group previously reported that HML-2 overexpression leads to neuronal death[18]. We hypothesized that the decrease in ASRGL1 caused by HML-2 overexpression and the subsequent effect on TDP-43 might be one underlying mechanism of HML-2-driven neurotoxicity. As expected, decreased viability was detected in HML-2 transfected iPSC-derived neurons compared to neurons transfected with a control vector. Co-transfection with ASRGL1 completely rescued the neurotoxicity caused by HML-2 (Fig. 8i, j).

### Levels of HML-2 inversely correlate with ASRGL1 in brain

Similar to previous reports[18], increased levels of HML-2 expression were found in ALS patients compared to controls validating these findings in a new and larger cohort (Fig. 8k). *HML-2* RNA levels showed an inverse correlation with the levels of ASRGL1 protein (Fig. 8l). This correlation was not strong, but still significant.

A graphic representation of the potential processes triggered by ASRGL1 described in this study is presented in Fig. 9. Our hypothesis is that formation of isoaspartates would produce kinks in the peptide chain of TDP-43, making the protein more prone to misfolding. ASRGL1 interacts with TDP-43 in the perinuclear region to cleave detrimental isoaspartates, as they would prevent substrate recognition by proteases[13,14] of the proteasome. Inhibition of ASRGL1 expression, by HML-2 RNA or cDNA silencing and/or other yet unknown mechanisms would cause accumulation of iso-aspartates in TDP-43, producing aberrant forms that cannot get degraded, leading to accumulation and eventually aggregation in the cytoplasm (Fig. 9).

## Discussion

Protein dysfunction and aggregation have been described in almost all neurodegenerative diseases. TDP-43 cytoplasmic aggregates are found in the brain of more than 97% of individuals with ALS[37]. Deamidation of asparagines and formation of isoaspartates is a degenerative protein modification which occurs by non-enzymatic processes, but their role in the pathophysiology of neurodegenerative diseases is largely unknown. In this study, we described the contribution of ASRGL1, an enzyme that degrades detrimental isoaspartyl peptides, to the

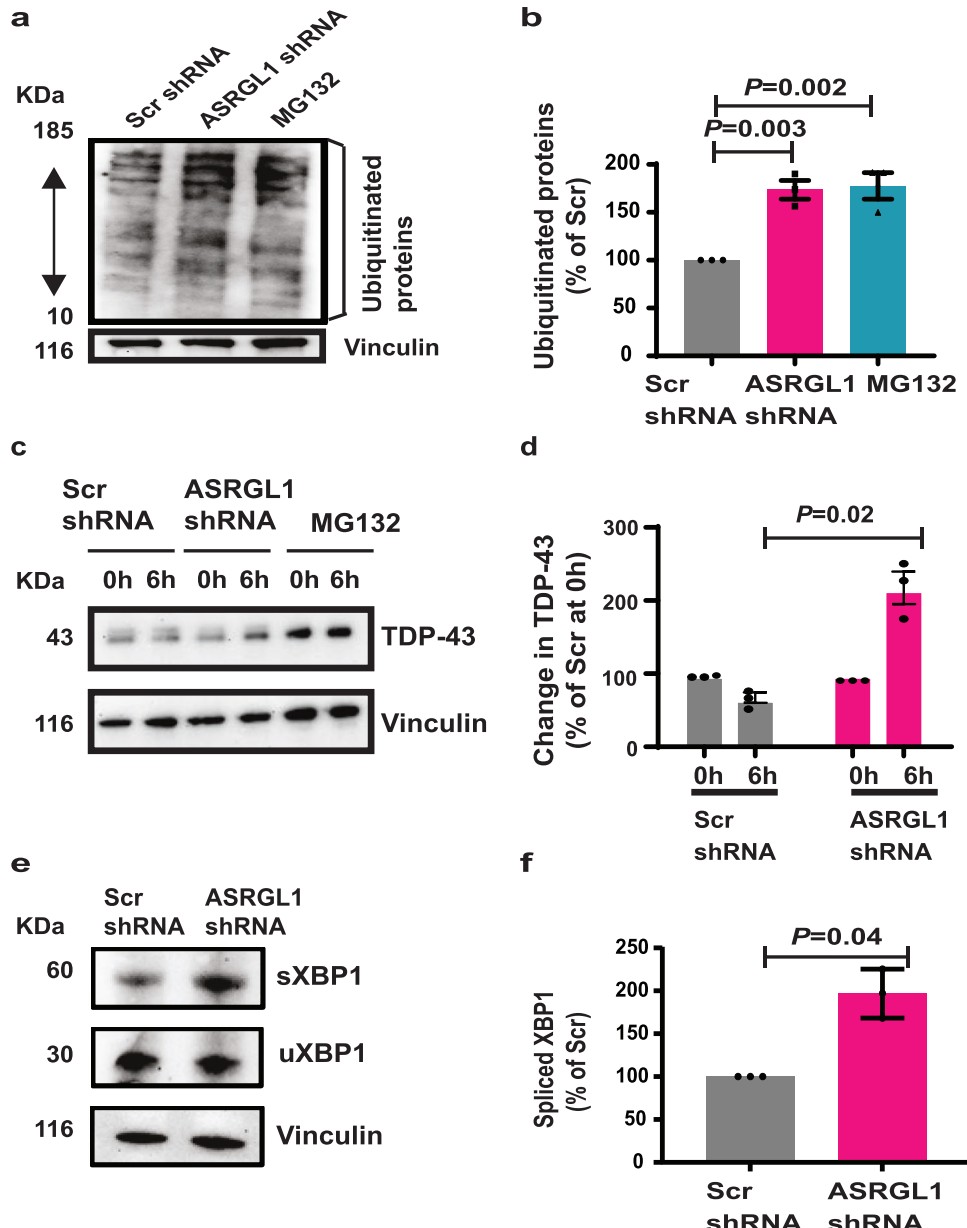

**Fig. 5 | ASRGL1 silencing impairs protein degradation. a, b** Neuronal cultures were transfected with ASRGL1 shRNAs or scrambled shRNAs. **a** Representative image of a western blot for ubiquitinated proteins. **b** Comparison of ubiquitinated proteins levels in percentage relative to the scrambled shRNA transfected cells (Brown-Forsythe ANOVA test with Bonferroni correction; Mean ± SEM; number of experimental replicates = 3)). **c, d** After transfection with scrambled shRNAs or ASRGL1 shRNAs, neuronal cultures were treated with cycloheximide (CHX) for 6 h. **c** Representative blot showing the levels of TDP-43 before and after 6 h of CHX.

**d** Percentage of variation in TDP-43 levels after 6 h of CHX (number of experimental replicates = 3; Unpaired $T$-test; Mean ± SEM). **e, f** Neuronal cultures were transfected with ASRGL1 shRNAs or scrambled shRNAs. **e** Western blot of spliced XBP1 (sXBP1), unspliced (uXBP1) and vinculin in ASRGL1-silenced and control neurons. **f** Comparison of sXBP1 levels in percentage relative to the scrambled shRNA transfected cells (Mann–Whitney test; Median (IQR); number of experimental replicates = 4). All pairwise comparisons in the Figure were performed with two-sided tests. Source data are provided as a Source Data file.

formation of TDP-43 proteinopathy and its implications for ALS pathogenesis. A striking observation was the highly significant decrease in *ASRGL1* expression in ALS individuals compared to controls by RNA seq in 391 brain samples ($p = 6.64E−9$). This decrease was confirmed by PCR and at the protein level by immunohistochemistry and western blotting. These differences were also evident in motor neurons derived from iPSCs lines. At a cellular level we found that the loss of ARGL1 was associated with cytoplasmic accumulation of TDP-43. Together, these observations suggest that ASRGL1 may play a critical role in maintaining TDP-43 homeostasis and may be a key contributor to the pathophysiology of ALS.

**Deamidation of TDP-43 leads to its accumulation in ALS**

We determined that TDP-43 has several asparagine residues that are naturally prone to deamidation, and consequently form isoaspartates. The C-terminal region of the protein was found to be most vulnerable to deamidation. As reported for other proteins[38], isoaspartates were predicted to produce conformational changes in TDP-43 that increase the free energy, and therefore decrease the stability of the protein[39]. Asparagine deamidation can modify the orientation relative to the rest of the polypeptide chain[40], altering the three-dimensional conformation of several human proteins inducing misfolding, and introduces a negative charge, promoting aggregation[12,38].

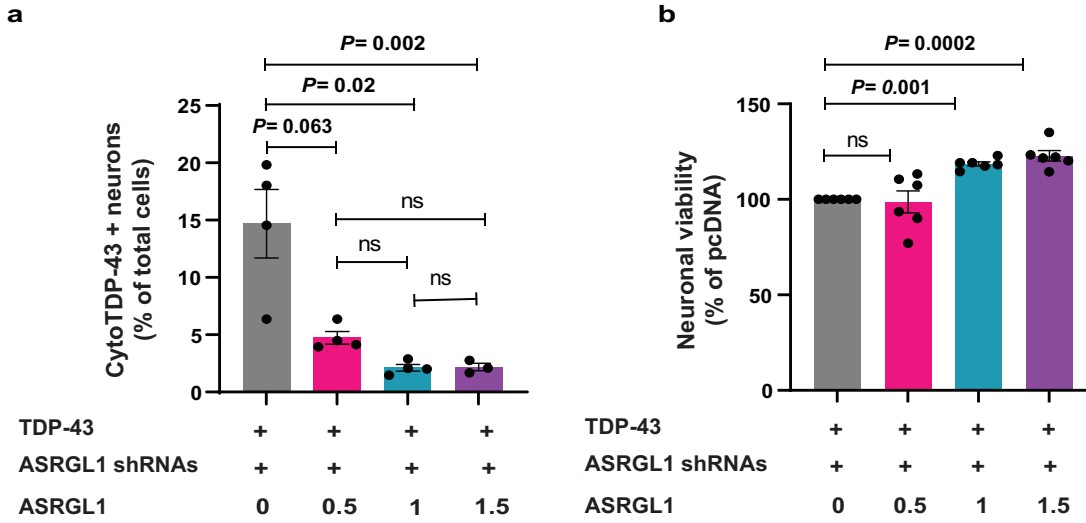

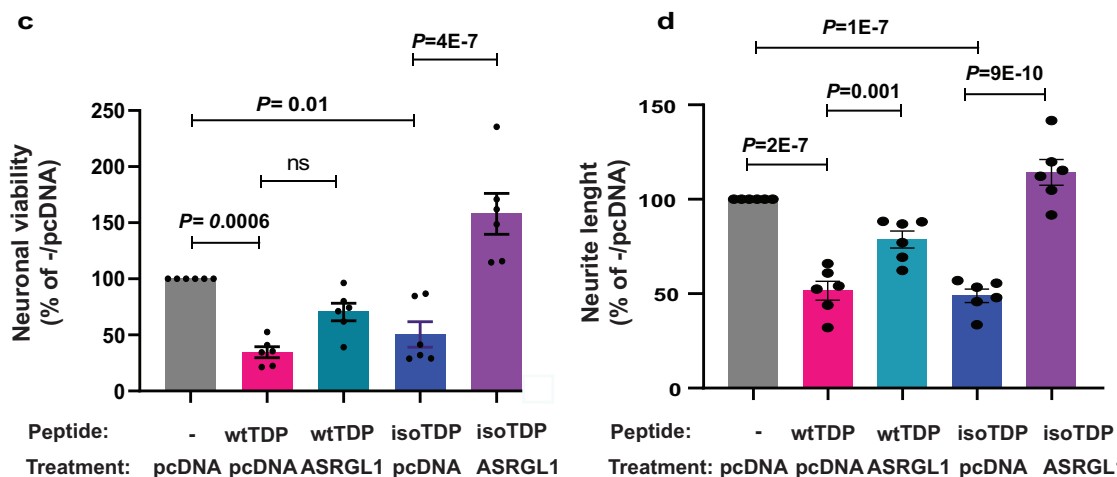

**Fig. 6 | ASRGL1 rescues TDP-43-proteinopathy and associated neurotoxicity.**
**a**, **b** iPSCs-derived human neuronal cultures from a normal donor were co-transfected with TDP-43, with shRNAs against ASRGL1 or scrambled shRNAs and with increasing concentrations of an ASRGL1-encoding plasmid or with pcDNA as a control. The number of cells presenting cytoplasmic TDP-43 was analyzed by immunofluorescence followed by confocal microscopy. **a** Percentage of neurons presenting cytoplasmic TDP-43 (one-way ANOVA with Bonferroni correction; Mean ± SEM; number of experimental replicates = 4). **b** Percentage of neuronal viability in relation to the control (pcDNA), as measured by flour spectrometry after applying Alamarblue (Thermo Fisher) (Brown-Forsythe ANOVA test with Bonferroni correction; Mean ± SEM; number of experimental replicates = 6). **c**, **d** IPSc-derived human neuronal cultures from a normal donor were co-transfected with wild type or isoaspartate containing C-terminal TDP-43 peptides, and with an ASRGL1 plasmid or an empty vector (pcDNA). **c** Percentage of neuronal viability in relation to the control (neurons transfected with pcDNA) (Brown-Forsythe ANOVA test with Bonferroni correction; Mean ± SEM; number of experimental replicates = 6). **d** Neurite length expressed as a percentage of the control (neurons transfected with pcDNA) (Brown-Forsythe ANOVA test with Bonferroni correction; Mean ± SEM; number of experimental replicates = 6). All pairwise comparisons in the Figure were performed with two-sided tests. Source data are provided as a Source Data file.

Deamidation of TDP-43 was detected more frequently in ALS patients than in controls. In a study that identified the post-translational modifications of TDP-43 in brain cytoplasmic inclusions of ALS patients by mass spectrometry, deamidation of asparagine residues was the most prevalent post-translational modification[41]. Deamidation of asparagine in voltage-dependent anion channel 1, a protein of the outer mitochondrial membrane known to interact with SOD1 mutants, has been identified in an ALS-related SOD1 mutant cell model[42]. Interestingly, most ALS-associated mutations of TDP-43 are found in the C-terminal region, where the asparagine residues predicted to deamidate are located[43].

The accumulation of isoaspartate residues is not unique to ALS. Isoaspartate residues in amyloid-beta peptides have been detected in Alzheimer's Disease (AD) where they increase toxicity and promote aggregation[44,45]. In vascular dementia there is deamidation of sodium-potassium transporting ATPase in residues close to the catalytic and regulatory sites of the protein[46] and in synapsin-1. The most studied enzyme involved in controlling the accumulation of iso-aspartates is isoaspartic acid methyltransferase (PIMT). PIMT catalyzes a methyl esterification reaction that initiates the conversion of these altered residues to the normal L-aspartyl form. Mice lacking this enzyme have elevated levels of damaged residues and die from

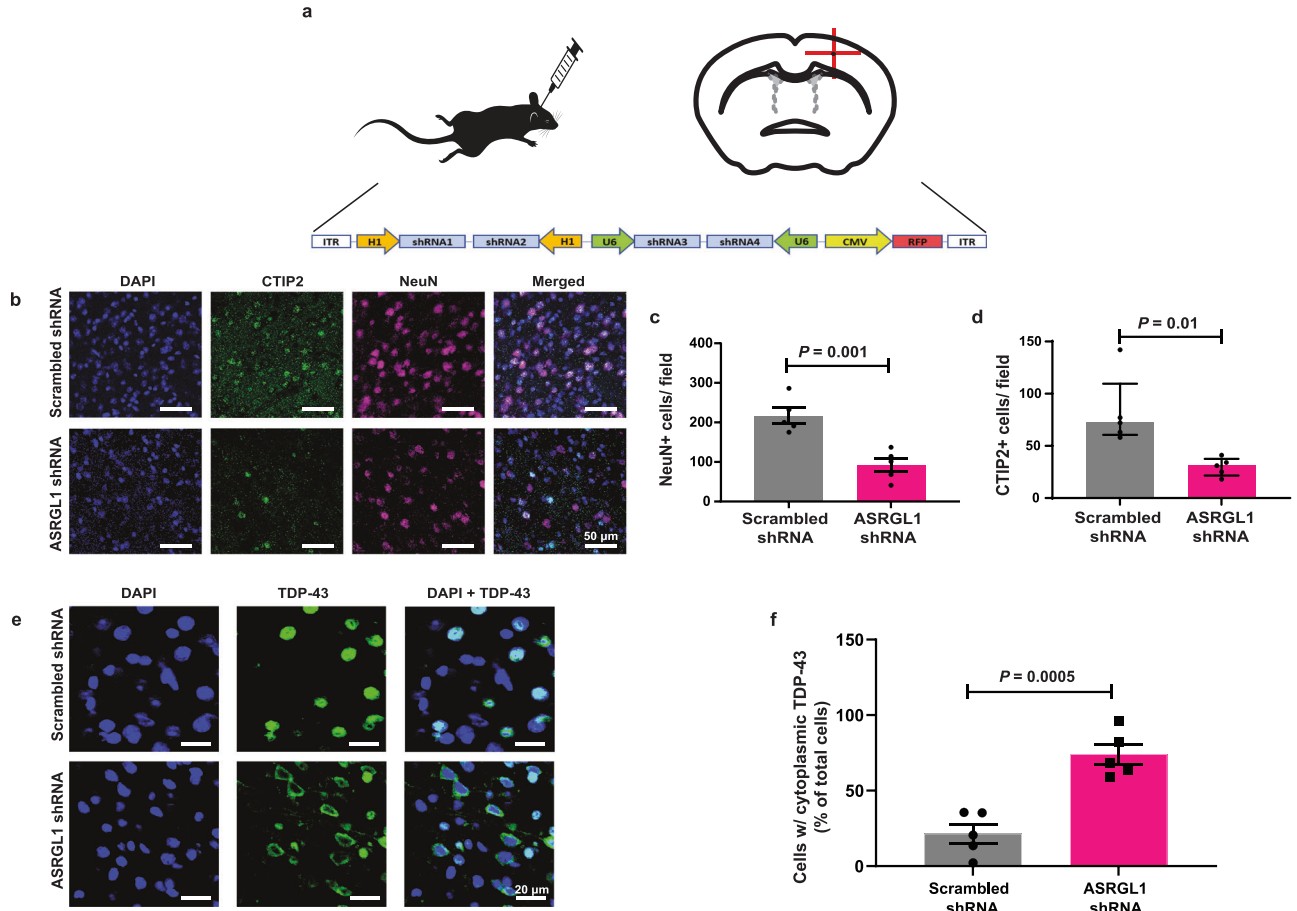

**Fig. 7 | ASRGL1 silencing triggers death of motor neurons and TDP-43 proteinopathy in mice.** Adult female C57BL/6 mice received a stereotaxic injection to deliver 2 μl of viral solution (10e13 AAV9 viral particles/ml) in the motor cortex. Half of the mice (*n* = 5) received particles with a construct encoding 4 scrambled shRNAs and the other half (*n* = 5) received particles with a construct encoding 4 shRNAs against *ASRGL1* (sequences in Supplementary Table 5). One month after the injection, brain sections were immunostained and analyzed by confocal microscopy. **a** Schematic representation of the injection site in the motor cortex (coordinates: 1.5 mm lateral, 1.1 mm anterior, 1.6 mm ventral) and the shRNA construct delivered within the AAV9 viral particles. **b** Representative images of brains sections from an ASRGL1-silenced mouse and a control stained for NeuN and CTIP2. **c** Comparison of NeuN+ cells (neurons) in the motor cortex of ASRGL1-silenced mice and controls (Unpaired *T*-test; Mean ± SEM). **d** Comparison of CTIP2+ cells (motor neurons) in the motor cortex of ASRGL1-silenced mice and controls (Mann–Whitney; Median (IQR)). **e** Brain section of an ASRGL1-silenced mouse showing cytoplasmic staining of TDP-43 (arrow heads), compared to a control mouse showing nuclear staining of TDP-43. **f** Percentage of cells showing cytoplasmic TDP-43 in ASRGL1-silenced mice and controls (Unpaired *T*-test; Mean ± SEM). All pairwise comparisons in the Figure were performed with two-sided tests. Source data are provided as a Source Data file.

tonic-clonic seizures[47]. However, even in the absence of PIMT, intracellular accumulation of damaged proteins is limited, suggesting that other pathways may be at play[48]. Since the discovery that ASRGL1 exhibits isoaspartyl peptidase activity, it has been proposed that it may act synergistically with PIMT to relieve accumulation of potentially toxic isoaspartyl peptides in mammalian brain and other tissues[15].

### Loss of ASRGL1 leads to TDP43 misfolding, cytoplasmic aggregation and malfunction

Several lines of evidence show that TDP-43 interacts with ASRGL1 and thus may be a substrate for this enzyme. Direct interactions between the proteins were shown by thermal shift assay and in neurons in vitro and in brain tissue by a proximity ligation assay. The two proteins could also be colocalized in the perinuclear regions in autopsy brain tissue. Loss of ASRGL1 in neurons led to an accumulation of misfolded, phosphorylated and fragmented TDP-43, suggesting that ASRGL1 is necessary for the physiological degradation of aberrant forms of TDP-43. This causes formation of cytoplasmic aggregates, as seen in neuronal cultures and in the mouse model,

following knocking down of ASRGL1. One consequence of the aggregation of TDP-43 in the cytoplasm is the loss of its nuclear functions. TDP-43 intervenes in the splicing of several genes, including UNC13A[29], which has a critical role in synaptic transmission[49]. We found that loss of ASRGL1 causes TDP-43 to malfunction, as determined by the increase in a cryptic exon in the mRNA and a decrease in the protein expression of UNC13A.

We have shown that ASRGL1 silencing causes an accumulation of ubiquitinated proteins, activates the UPR and increases the half-life of TDP-43. This indicates an impairment of protein degradation. Misfolded and dysfunctional proteins are degraded by the proteasome/autophagosome cellular systems. Proteostasis deficiency is one of the key features of ALS and other neurodegenerative disorders and might be responsible for protein accumulation and aggregation. In fact, ALS-associated mutations have been found in genes implicated in proteasome degradation, ER-associated protein degradation (ERAD) and autophagy[50]. Inhibition of the ubiquitin-proteasome system by treatment with epoxomicin or MG132 was associated with cytoplasmic TDP-43 fragments in both immortalized neuronal cell lines and primary hippocampal and cortical cultures[51]. Overexpression of TDP-43 also

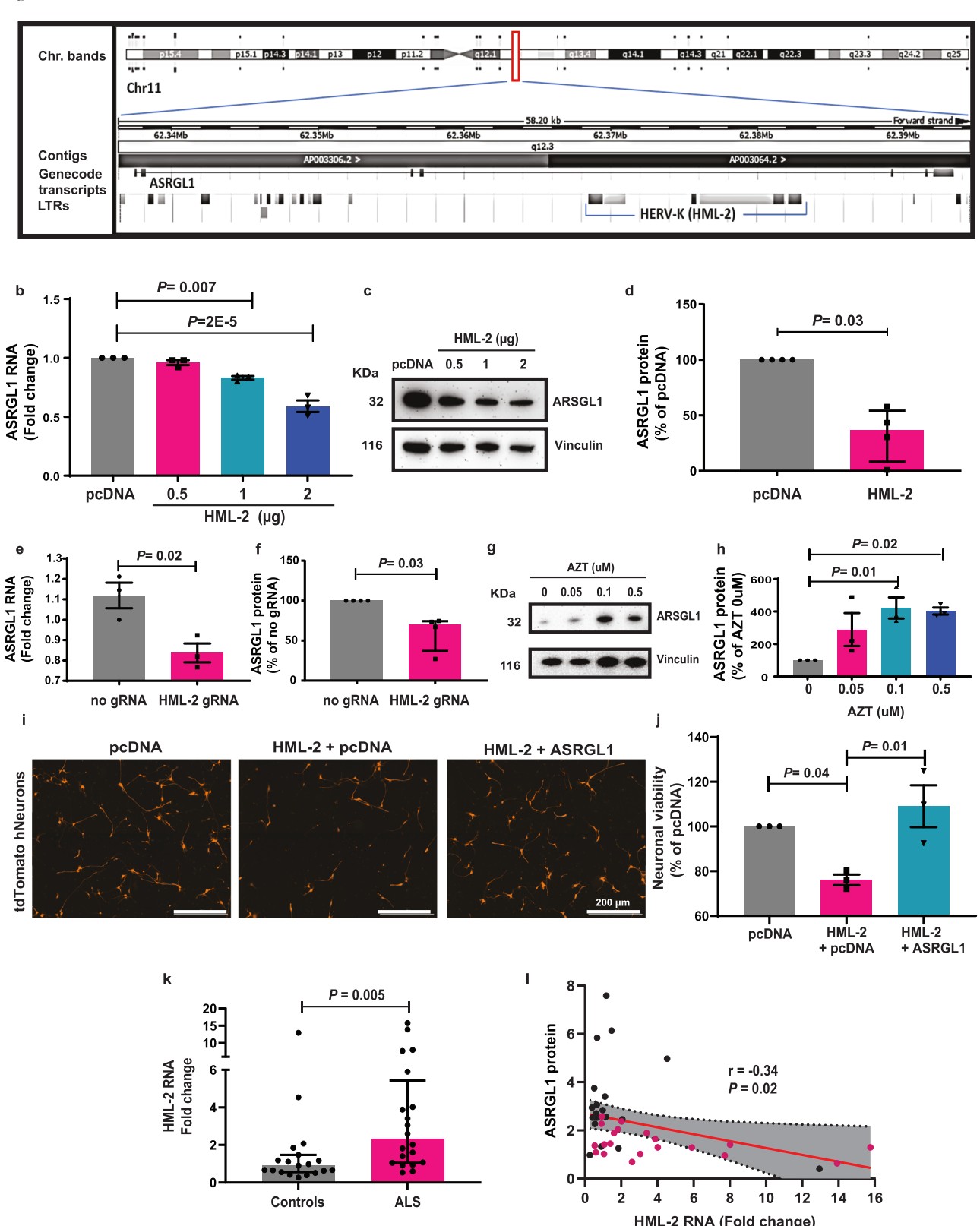

impairs the proteasome degradation pathway, suggesting that interference with the ubiquitin-proteasome system is involved in TDP-43 associated toxicity[52]. One of the main consequences of proteasome dysfunction or inhibition is the activation of ER-stress pathways and UPR, which occurs in familial and sporadic ALS, as determined by analysis of post-mortem tissues[53].

**Activation of HML-2 might cause ASRGL1 down regulation**

An important question is the mechanism by which ASRGL1 is down regulated in ALS. Interestingly, a copy of *HML-2* is located in the opposite strand of the *ASRGL1* gene thus, the RNA of *HML-2* is complementary to the RNA of *ASRGL1*. Similar to previous studies with several other long non-coding RNAs (lncRNAs)[54], lncRNA HERV

**Fig. 8 | ASRGL1 is downregulated by HML-2 and rescues HML-2-driven neurotoxicity. a** Diagram showing the location of an HML-2 intronic copy in ASRGL1 gene (Ensembl.org). **b**–**d** iPSCs-derived human neuronal cultures from a normal donor were transfected with an HML-2 plasmid or with an empty vector (pcDNA). **b** *ASRGL1* RNA levels quantified by real-time qPCR (Brown-Forsythe ANOVA test with Bonferroni correction; Mean ± SEM; number of experimental replicates = 3). **c** Western blot for ASRGL1 and vinculin (loading control). **d** Comparison of ASRGL1 protein levels expressed as a percentage of the control (pcDNA) in neuronal cultures transfected with pcDNA (1 µg) or HML-2 (1 µg), by western blotting (number of experimental replicates = 4; Mann–Whitney test; Median (IQR)). **e**, **f** IPSc-derived human neuronal cultures from a normal donor were transfected with a CRISPR/dCAS9 construct with guide-RNAs (gRNAs) targeting the LTR to activate endogenous expression of *HML-2* or without gRNAs. **e** *ASRGL1* RNA levels, by qPCR (Unpaired *T*-test; Mean ± SEM). **f** Comparison of ASRGL1 protein levels expressed as a percentage of the control (no guide RNAs), by western blotting (number of experimental replicates = 4; Mann–Whitney test; Median (IQR)). **g**, **h** iPSCs-derived neural stem cells were treated with increasing concentrations of zidovudine (AZT). **g** Western blot for ASRGL1 and vinculin. **h** ASRGL1 protein levels expressed as a percentage of the control (no zidovudine) (Brown-Forsythe ANOVA test with

Bonferroni correction; Mean ± SEM; number of experimental replicates = 3). **i**, **j** IPSc-derived human neuronal cultures from a normal donor stably expressing td-Tomato protein were transfected with a pcDNA plasmid, co-transfected with pcDNA and HML-2 or co-transfected with HML-2 and a plasmid encoding ASRGL1, and neuronal viability was measured by automated fluorescence microscopy. **i** Representative images of fluorescence microscopy. **j** Percentage of neuronal viability relative to the control (pcDNA transfected cells) (Brown-Forsythe ANOVA test with Bonferroni correction; Mean ± SEM; number of experimental replicates = 3). **k**, **l** RNA and proteins were extracted from brain cortex samples of ALS patients (*n* = 20) and normal controls (*n* = 19); the levels of HML-2 env RNA were analyzed by qPCR and the levels of ASRGL1 protein were analyzed by western blotting. **k** HML-2 RNA levels in ALS brains and normal controls, by qPCR (Mann–Whitney test; Median (IQR)). **l** Correlation between *HML-2* RNA and ASRGL1 protein levels (ratio of ASRGL1 to Vinculin, as measured by western blotting) (Pearson's *r* test; red dots: ALS; black dots: controls). The levels of ASRGL1 (ratio of ASRGL1 to Vinculin) in ALS and control brains are shown in Fig. 1f–h. All pairwise comparisons in the Figure were performed with two-sided tests. Source data are provided as a Source Data file.

transcripts might also affect host gene expression. We hypothesized that *HML-2* RNA could bind to the pre-mRNA of *ASRGL1* in the nucleus interfering with its splicing and maturation. In a recent study, where the impact of surrounding genes on intronic *HML-2* provirus expression was investigated, only the expression of the *HML-2* provirus at chromosome 11q12.3 located in the intron of ASRGL1 significantly correlated with that of its neighboring gene (*ASRGL1*), which makes it a likely candidate for read-through transcription[36]. This might be caused by an intron retention, an alternative splicing mode whereby introns, rather than being spliced out as usual, are retained in mature mRNAs. This supports the possibility that the antisense silencing effect of HML-2 on ASRGL1 could also occur at the level of the mature *ASRGL1* RNA on the intronic sequence, interfering with its translation and/or activating the RNA-induced silencing complex[55]. Alternatively, if *HML-2* is activated, its RNA can get reverse transcribed into cDNA, which could also potentially bind to the *ASRGL1* RNA. Moreover, since TDP-43 is involved in RNA processing, there is the possibility of a pathogenic loop whereby TDP-43 dysfunction causes a decrease in ASRGL1 expression by affecting its splicing, which would enhance the TDP-43 proteinopathy. Moreover, considering the RNAseq results presented here, where a large cohort of ALS and controls was analyzed, *ASRGL1* loss seems to be a more prevalent feature of ALS than HML-2 overexpression, which was found in a subset of ALS patients (20–30%). Therefore, ASRGL1 depletion might be a common consequence of several causative factors besides HML-2, such as mutations in the *ASRGL1* gene or in its transcription factors. Future studies are required to elucidate the exact mechanisms of silencing. Several lines of evidence support the possibility of HML-2 silencing ASRGL1 expression. Levels of *HML-2* transcripts have been shown to be elevated in brains of individuals with ALS[17,18,56], and we found that levels of ASRGL1 inversely corelated with *HML-2* in brain. Further, forced expression of HML-2 by transfection or induction of the endogenous HML-2 resulted in a decrease in ASRGL-1 in a dose-responsive manner. Moreover, AZT, which is known to inhibit HML-2[57], increased the levels of ASRGL1.

Several studies support a complex relationship between ASRGL1, TDP-43 and transposable elements. For example, *ASRGL1* missense mutations have been associated with late-onset AD[58] and with retinal degeneration[59]. The relationship between proteinopathies and transposable elements might also be bi-directional. An increase in the levels of endogenous retroviruses (ERVs), including gypsy, in response to TDP-43 proteinopathy has been reported in a human TDP-43-Drosophila model and activation of ERVs in glial cells by TDP-43 leads to DNA damage and death of adjacent neurons[60]. Transposable

elements dysregulation is also a mediator of neuronal death in tauopathies[61]. Expression of either Drosophila mdg4-ERV (gypsy) or the human ERV, HERV-K (HML-2) are each sufficient to stimulate cytoplasmic aggregation of human TDP-43[62]. TDP-43 has been shown to bind to *HML-2* nucleic acids[18]; although a recent paper has shown that TDP-43 does not activate HML-2 expression, it might possibly favor its accumulation[63], which could lead to inhibition of ASRGL1. Therefore, a vicious loop could be triggered where increased levels of HML-2 can inhibit ASRGL1 leading to TDP-43 accumulation, which could further increase HML-2 levels.

Our results point to a neuroprotective effect of ASRGL1, since ASRGL1 rescues the neurotoxicity caused by full TDP-43 overexpression, by TDP-43 C-terminal peptides and by HML-2. Previously our group had shown that overexpression of HML-2 caused cell death and decreased neurite length in neuronal cultures[18]. This suggests that inhibition of ASRGL1 production might be one of the mechanisms of HML-2-driven neurotoxicity.

## ASRGL1 interacts with several proteins critical to ALS pathophysiology

To explore other ASRGL1 interacting proteins, we performed a search in BioGRID (https://thebiogrid.org/). Interestingly, ASRGL1 interacts with several other proteins associated with ALS and other neurodegenerative diseases (Supplementary Fig. 14). It interacts with Glycogen Synthase Kinase 3 β (GSK3β), a protein- kinase involved in tau phosphorylation. GSK3β is elevated in the brain of ALS individuals[64]. Several GSK3β inhibitors are currently under investigation in preclinical and clinical studies to treat neurodegenerative diseases. ERBB2 is a receptor for Neuregulin-1, which is elevated in the spinal cord of ALS individuals and SOD1 mice and might underlie the microglial activation seen in the disease[65]. Stathmin-2 (STMN2), another ASRGL1 interactor, is an essential protein for axonal growth that has been recently associated with ALS. Truncated forms of STMN2 are elevated in the brain of patients with FTD and is a marker for TDP-43 dysfunction[66]. At least two Mitogen-activated protein kinase (MAP kinases) have been found to interact with ASRGL1 (MAPK81P2 and MAPK2K5). MAPK pathway is a fundamental mitogen/stress-activated signal transduction pathway that is altered in ALS[67]. Finally, another ASRGL1 interactor also related with neurodegeneration is the prion protein (PRNP). Dysfunction or loss of ASRGL1 might be a common link between several neurodegenerative diseases, whose pathogenic aggregative proteins seem to act like prions[68]. Studies to specifically evaluate the biological significance of the interaction of ASRGL1 and the above-mentioned proteins in the context of ALS would be necessary.

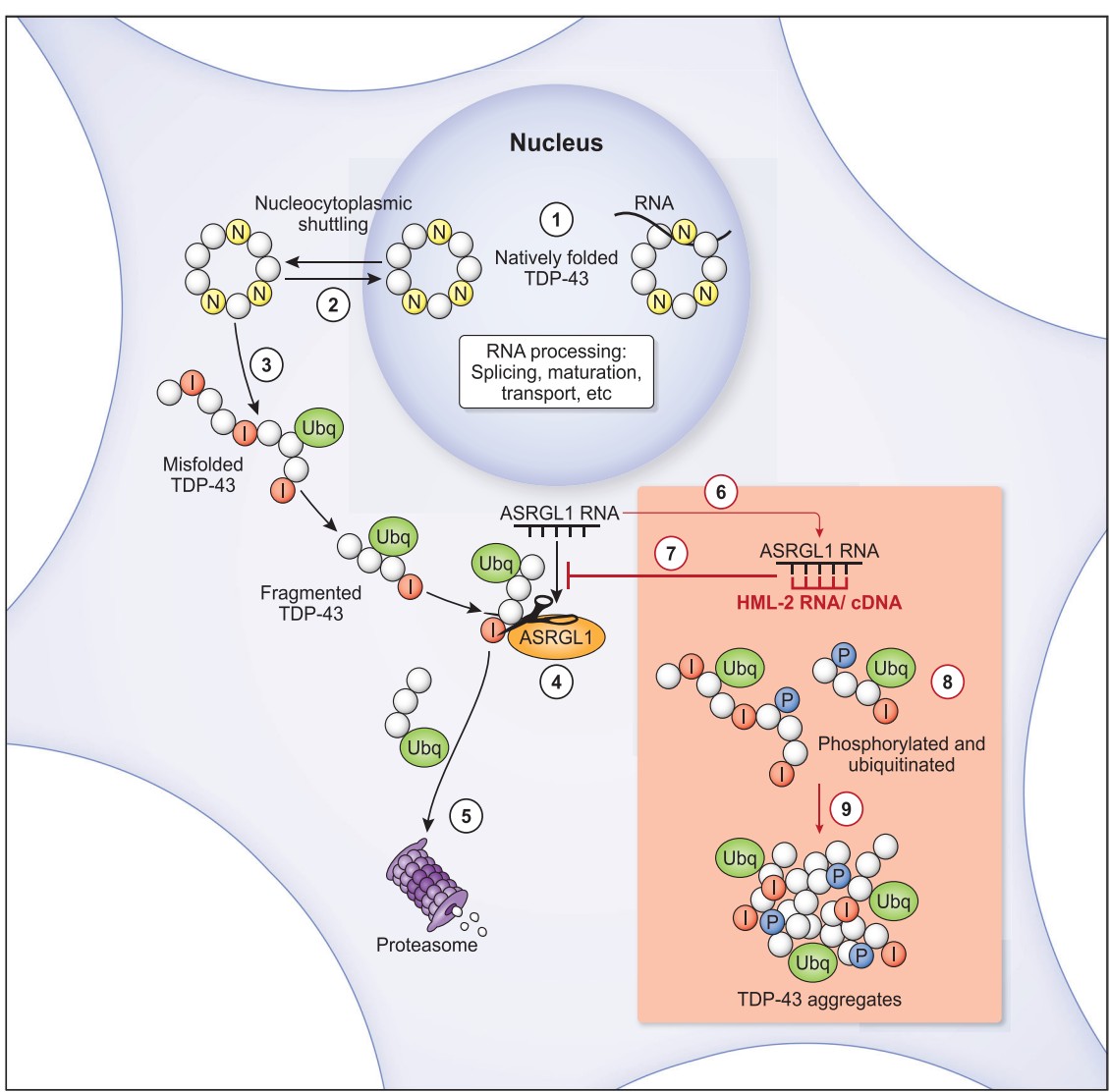

**Fig. 9 | Schematic representation of potential processes triggered by loss of ASRGL1 resulting in TDP-43 proteinopathy in ALS.** (1) TDP-43 is involved in RNA transcription, splicing, stability, transport, and translation. (2) It is predominantly found in the cell nucleus but to perform its functions, TDP-43 is shuttled in its native form from the nucleus to the cytoplasm. (3) Isoaspartates (I) are formed spontaneously and non-enzymatically by deamidation of asparagine residues (N) of TDP-43. They introduce a kink in the peptide chain, favoring misfolding. Misfolded and fragmented TDP-43 are ubiquitinated for proteasome degradation. (4) Since proteases do not recognize isoaspartates residues, they must be cleaved from the peptide chain by ASRGL1 in the perinuclear region for the misfolded/fragmented protein to be (5) degraded by the proteasome. (6) The *HML-2* copy is in the opposite strand of the *ASRGL1* gene thus, the RNA of *HML-2* is complementary to the RNA of *ASRGL1* (*HML-2* cDNA could also base pair with the *ASRGL1* RNA). (7) HML-2 sequences could cause an antisense silencing on ASRGL1, interfering with its splicing, translation and/or activating the RNA-induced silencing complex. (8) If ASRGL1 is inactivated or depleted, isoaspartates residues cannot get cleaved and the aberrant TDP-43 forms (hyper-phosphorylated, misfolded and fragmented peptides) do not get degraded, (9) leading to a toxic aggregation in the cytoplasm. (Black arrows indicate physiological processes and red arrows indicate pathological processes).

## Study limitations

Our study has several limitations. Part of the deamidation of asparagines found in TDP-43 in the brain samples might be produced by the sample processing; however, since all the samples were processed in the same way, the fact that deamidation was found more frequently in ALS patients than in controls support deamidation of TDP-43 as a process naturally occurring in vivo and associated with the disease. Besides, autopsy tissue represents end-stage disease. It remains to be determined when in the course of the illness, these cascades get initiated. It is unknown if these findings are also relevant to the anterior horn cells in the spinal cord although we have previously shown that HML-2 is activated in these cells[18]. This study was limited to sporadic form of ALS and hence, the relevance for familial forms of ALS needs to be determined. Despite these limitations, our findings provide mechanistic insight into ALS pathophysiology and possibly other neurodegenerative diseases. Our findings have several therapeutic implications. Gene therapy for the restoration of ASRGL1 enzymatic activity in the brain could be considered. ASRGL1 enzyme is also present extracellularly[69], thus replacement therapies by delivery of the enzyme intrathecally could be explored. Gene therapies targeting HML-2 such as antisense oligonucleotides, siRNA or shRNA could also be developed. Antiretroviral drugs that target HML-2 may have therapeutic potential as well. A pilot study using antiretroviral drugs has already been conducted in ALS patients which showed a decrease in HML-2 levels in blood[20] and an improved clinical response in those patients who experienced a decrease in HML-2 levels[19,31]. However, drugs that specifically target HML-2 need to be developed.

In conclusion, we demonstrate the importance of ASRGL1 depletion in the formation of TDP-43 proteinopathy. The observed loss of ASRGL1 in the brain of ALS patients supports its role in this devastating disease. The downregulation of ASRGL1 by HML-2 points to a possible pathogenic mechanism of this endogenous retrovirus in ALS. Restoration of this enzyme function may be a therapeutic approach in ALS and possibly in other disorders where similar TDP-43 pathology has been described, such as frontotemporal dementia, limbic-predominant age-related TDP-43 encephalopathy (LATE) and some cases of Alzheimer's disease.

## Methods

### RNA sequencing analysis

**Samples.** RNA sequencing data was acquired from the New York Genome Center (NYGC) ALS Consortium Database (https://www.nygenome.org/contact/) following institution's approval and Material Transfer Agreement. Demographic characteristics of the ALS patients and controls are described in Supplementary Table 1. All samples processed and analyzed by the NYGC were obtained with informed consent and following the regulations of the Ethics Committees and with approval of the IRBs of the participating institutions.

Our cohort comprised motor cortex samples from 307 sporadic ALS individuals with no known mutation and 16 ALS individuals with known mutations. There were 45 patients with a family history of ALS, 2 of whom had known mutations. The cohort also contained 68 unaffected individuals, only two of which had known mutations in ALS associated genes, and only one unaffected individual had a family history of ALS. We had 11 female and 15 male familial ALS patients, 13 female and 11 male sporadic ALS patients and 8 female and 6 male individuals with dementia and no ALS with $C9Orf72$ mutations in our cohort.

### Library preparation

Library preparation was performed as described elsewhere[70]. Briefly, RNA was extracted from flash-frozen patient samples homogenized in Trizol (15596026, Thermo Fisher Scientific, Waltham, MA, USA) -Chloroform and purified using QIAGEN RNeasy Mini kit (74104, QIAGEN, Germantown, MD, USA). RNA quality was assessed using a Bioanalyzer (G2939BA, Agilent, Santa Clara, CA, USA). RNA-seq libraries were prepared from 500 ng of total RNA using the KAPA Stranded RNA-seq kit with RiboErase (07962304001, Kapa Biosystems, Wilmington, MA, USA) for rRNA depletion and Illumina compatible indexes (NEXTflex RNA-seq Barcodes, NOVA-512915, PerkinElmer, Waltham, MA, USA).

### Sequencing and analysis

Pooled libraries (average insert size: 375 bp) were sequenced on an Illumina HiSeq 2500 using a paired end 125 nucleotide setting, to yield 40–50 million reads per library.

All samples were sequenced as paired-end reads and the strand of origin for each read was also identified. Samples that were sequenced on multiple lanes were combined. Trimmomatic[71] was used to trim the original sequencing files obtained from NYGC to 80 base pairs (bp) in order to remove barcodes and improve sample quality.

To account for differences in sample sequencing depth, all files were down sampled to 25 million reads. The resulting files were aligned to the hg38 reference genome (https://www.ncbi.nlm.nih.gov/assembly/GCF_000001405.26/) with the Spliced Transcripts Alignment to a Reference (STAR) software[72]. All resulting files for each sample were combined into a single count matrix and used for the following analyses.

### Differential expression analysis (DEA)

Samples in the count matrix file that were collected from the motor cortex were selected. Next, genes with a low number of transcript counts (less than 6) were removed as well as those that were significantly associated with biological sex (DESeq2 FDR adjusted $p$ value < 0.05). The total number of patient samples was 323 and the total number of control samples was 68. DEA was performed on the resulting non-normalized count data matrix using the DESeq2 package in R to compare the expression profiles of sporadic ALS patients and controls. Differentially expressed genes (DEGs) were counted as genes with a False discovery rate (FDR) adjusted $p$ value < 0.05. Unless indicated otherwise, all adjusted $p$ values were corrected using False discovery rate (FDR).

### Single-cell RNA sequencing analysis

Single-cell RNA sequencing data from a study of Dr Manolis and Dr Heiman's groups of the Massachusetts Institute of Technology (MIT) (Pineda et al.[21]) was secondary analyzed in collaboration with this group to compare the expression of $ASRGL1$ and $TARDBP$ between ALS individuals and controls (FTLD and non-neurological) in different subtypes of neurons and glia, in motor cortex (BA4) and in dorsolateral prefrontal cortex (BA9).

### Human sample analysis and selection for single-cell sequencing

Frozen postmortem brain samples were obtained from the Mayo Clinic Neuropathology Laboratory (Jacksonville, FL). All cases underwent standard neuroanatomic sampling by a single neuropathologist who performed the cutting of all autopsied procured brains (Mayo Clinic IRB 15-009452). We included a total of 73 age- and sex-matched individuals with sporadic ALS (sALS; $n = 17$), C9orf72-associated ALS (C9ALS; $n = 16$), sporadic FTLD (sFTLD; $n = 13$), C9orf72-associated FTLD (C9FTLD; $n = 11$) or found pathologically normal (PN; $n = 16$). All ALS and FTLD samples selected showed TDP-43 pathology. C9orf72 ALS was confirmed by P62 immunohistochemistry and southern blotting of repeat expansions of the C9orf72.

### Human sample preparation

For the single-nucleus study BA4 (primary motor cortex) was sampled from frozen tissue stored at −80 °C, as it is the most affected area in ALS, and BA9 (dorsolateral prefrontal) was sampled as a control region.

### Isolation of nuclei from post-mortem frozen human brain tissue for single nuclear RNA-sequencing

All procedures were performed on ice. Each piece of tissue was homogenized in 700 µl of homogenization buffer (320 mM sucrose, 5 mM CaCl2, 3 mM Mg(CH3COO)2, 10 mM Tris HCl [pH 7.8], 0.1 mM EDTA [pH 8.0], 0.1% NP-40, 1 mM β-mercaptoethanol, and 0.4 U/µl SUPERaseIn RNase Inhibitor (Thermo Fisher Scientific, Waltham MA) with a 2 ml KIMBLE Dounce tissue grinder (MilliporeSigma, Burlington MA) using 10 strokes with loose pestle followed by 10 strokes with tight pestle. From the human tissue samples 100 mg of gray matter was sectioned and homogenized. Homogenized tissue was filtered through a 40 µm cell strainer and mixed with 450 µl of working solution (50% OptiPrep density gradient medium (MilliporeSigma, Burlington MA), 5 mM CaCl2, 3 mM Mg(CH3COO)$_2$, 10 mM Tris HCl [pH 7.8], 0.1 mM EDTA [pH 8.0], and 1 mM β-mercaptoethanol). The mixture was then slowly pipetted onto the top of an OptiPrep density gradient containing 750 µl of 30% OptiPrep Solution (134 mM sucrose, 5 mM CaCl2, 3 mM Mg(CH3COO)2, 10 mM Tris HCl [pH 7.8], 0.1 mM EDTA [pH 8.0], 1 mM β-mercaptoethanol, 0.04% NP-40, and 0.17 U/µl SUPERase In RNase Inhibitor) on top of 300 µl of 40% OptiPrep Solution (96 mM sucrose, 5 mM CaCl$_2$, 3 mM Mg(CH3COO)$_2$, 10 mM Tris HCl [pH 7.8], 0.1 mM EDTA [pH 8.0], 1 mM β-mercaptoethanol, 0.03% NP-40, and 0.12 U/µL SUPERase In RNase Inhibitor) inside a Sorenson Dolphin microcentrifuge tube (MilliporeSigma, Burlington MA). Nuclei were pelleted at the interface of the OptiPrep density gradient by centrifugation at $10,000 \times g$

for 5 min at 4 °C using a fixed angle rotor (FA-45-24-11-Kit). The nuclear pellet was collected by aspirating ~100 µl from the interface and transferring to a 2.5 ml Eppendorf tube. The pellet was washed with 2% BSA (in 1x PBS) containing 0.12 U/µl SUPERase In RNase Inhibitor. The nuclei were pelleted by centrifugation at $300 \times g$ for 3 min at 4 °C using a swing-bucket rotor (S-24-11-AT). Nuclei were washed three times with 2% BSA and centrifuged under the same conditions. The nuclear pellet was resuspended in 100 µl of 2% BSA. Droplet-based snRNA sequencing libraries were prepared using the Chromium Single Cell 3′ Reagent Kits v3 and v3.1 (10x Genomics, Pleasanton CA) according to the manufacturer's protocol and sequenced on a NovaSeq 6000 at the Broad Institute Genomics Platform.

### Sequencing data pre-processing
FASTQ files were aligned to the human reference genome GRCh38 allowing mapping to intronic regions. Cell Ranger v6.0 (10x Genomics, Pleasanton CA) was used for genome alignment and feature-barcode matrix generation.

### Clustering and cell type annotation
We used the ACTIONet R package for data cleaning, batch correction, clustering, imputation, and cell type annotation as with some modifications. In brief, raw count matrices were prefiltered by removing barcodes containing less than 700 unique genes or greater than 20% mitochondrial RNA content. Counts were depth-normalized across samples and log-transformed using the batchelor R package. Further filtering was performed by iteratively removing cells with abnormally low or high RNA content (relative to the distribution of its specific cluster), ambiguous overlapping profiles resembling dissimilar cell types (generally corresponding to doublet nuclei), and cells corresponding to graph nodes with a low k-core or low centrality in the cell-cell similarity network (generally corresponding to high ambient RNA content or doublet nuclei). We also filtered cells overexpressing immediate early genes (e.g., FOS, JUN, JUNB, JUND) not correlated with phenotype, and indicative of hypoxia or cell damage. These comprised a negligible fraction of filtered cells.

Following this rigorous filtering regime, we retained 625,973 unique cellular profiles corresponding to ~8500 post-quality control nuclei per donor, with an average of ~14,000 unique transcripts, ~4200 unique genes, and mitochondrial RNA content of 1.48% per nucleus.

A curated set of known major cell type markers was used to annotate individual cells with their expected cell type and assign a confidence score to each annotation. For ill-characterized or ambiguous subpopulations, we assigned identity based on visually distinct, well-defined gene expression domains reproducibly identified by Leiden and archetypal analysis clustering at multiple resolutions. Descriptive labels were determined by the unique co-expression of marker genes identified by Wilcoxon rank sum test for each cluster relative to the background.

### Differential gene expression analysis
We employed a pseudo bulk approach considering each donor individual as a single biological replicate. For the coarse analysis, neuronal subtypes were aggregated into excitatory and inhibitory classes. For the granular, cell type-specific analysis, rare subclusters that lacked sufficient representation were merged if they were similar enough to still reasonably be classified as one cell type. More specifically, rare clusters were merged if their subclusters were directly adjacent and belonged to the same supercluster (e.g., subclusters of PV neurons or the two L5 VAT1L+ clusters) and could not be analyzed independently. We required that a cluster be represented across at least 4 disease and 4 control individuals, separately for each disease and brain region, and with at least 20 cells per individual. If this criterion could not be met

and rare clusters could not be merged with a larger group, these clusters were dropped from the analysis.

For each differential expression cluster, only genes present in at least 10% of cells were retained for analysis. Counts were depth normalized, scaled by library-size, and log-transformed. Pseudo bulk expression profiles were computed by averaging normalized log-counts within each cluster for each unique sample. We used surrogate variable analysis to identify and remove sources of unknown variance. Cell type-specific pseudo-bulk differential gene expression (DGE) analysis was performed with the limma R package using age, sex, and disease group as design covariates.

Genes were considered differentially expressed if they had a log2-fold change >1 standard deviation with an FDR-corrected $p$ value < 0.05. To ensure these results were robust and confirm that our chosen thresholds were reasonable, we performed permutation testing whereby we randomly swapped covariate labels and repeated the pseudobulk analysis. This analysis estimated the number of genes that would be classified as differentially expressed by chance (i.e., false positives). In all cases, zero or close to zero DEGs were detected.

### Cross-region comparisons
For each PN donor individual, per brain region, pseudo bulked gene expression profiles for each transcriptional subtype were generated by averaging log-normalized counts. Then the mean expression of the pseudo bulked profiles was used to compute the pairwise Pearson's correlation of transcriptome-wide gene expression across subtypes of MCX and PFC in human. To minimize the contribution of ubiquitous housekeeping genes and to amplify the contribution of cell type-specific gene expression, we regressed out the mean gene expression profile across all subtypes from the expression matrices before computing the correlation.

### Clinical samples for ASRGL1 and TDP-43 detection
Frozen brain samples (frontal cortex; Brodmann Area 6) from 20 sporadic ALS and 20 age and sex-matched controls were studied. We also analyzed formalin-fixed paraffin-embedded (FFPE) brain tissue (pre-motor cortex; BA6 and visual cortex; BA17) rom 16 individuals (4 sporadic ALS patients, 4 MS patients, 4 AD, 4 normal controls). Brain samples were kindly provided by the institutions described in Supplementary Table 2 following institutions' approvals and Material Transfer Agreements. Patient characteristics and sources of the samples are described in Supplementary Table 2.

### Chromogenic immunohistochemistry for ASRGL1 and TDP-43
Immunohistochemical analysis was performed on post-mortem brain tissues. Diaminobenzidine (DAB) staining and alkaline phosphatase (AP) staining were carried out for ASRGL1 and TDP-43, respectively. Brain samples (Supplementary Table 2) were fixed in 10% neutral buffered formalin (NBF) and embedded in paraffin. Formalin-fixed, paraffin-embedded (FFPE) blocks were sectioned at 5 µm and mounted on Super-Frost® Plus (Thermo Fisher Scientific). Immunohistochemical staining was performed on a Leica Bond-MAX automated slide stainer (Leica Biosystems) using protocols from the manufacturer. Rabbit polyclonal anti-ASRGL1 (1:250, Atlas Antibodies, #HPA055572) and mouse monoclonal anti-TDP-43 (1:4000, EnCor, #MCA-3H8) antibodies were used to label ASRGL1 and TDP-43, respectively. Images were processed using a whole slide scanner, Aperio (Leica Biosystems). Ten regions per slide were randomly selected. Each region area was 0.4 mm². Quantification of ASRGL1-positive cells per region was performed automatically using Gen5 Image Prime software (BioTek, Winooski, VT). Quantification of cells with cytoplasmic TDP-43 was done manually. All quantification was done in blind.

## Induced pluripotent stem cells

Induced pluripotent stem cells (iPSCs) lines were generated from deidentified PBMCs from 6 healthy human donors (Supplementary Table 3) and 6 sporadic ALS patients using a Sendai virus method, as previously described[73]. The lines were generated by the National Heart, Lung, and Blood Institute iPSCs core faculty at the National Institutes of Health (NIH). Samples were obtained under protocols 15-N-0125 (ALS patients) and 03-AG-0325 (Normal controls) following approval by the Institutional Review Board of NIH.

## Differentiation of motor neurons

Induced motor neurons (iMN) were differentiated from pluripotent stem cells following a protocol similar to that of Coyne et al.[74]. Briefly, iPSCs were seeded on a Matrigel-coated 6 well plate in E8 Flex medium (Thermo Fisher Scientific). The cells were seeded as aggregated spheres to get better motor neuronal differentiation. When ready for induction, the E8 Flex medium was replaced with Stage 1 iMN media for 6 days. Stage 1 iMN media contained Iscove Modified Dulbecco Media (IMDM) (Gibco) and Ham's F12 (Gibco) media mixed in a 1:1 ratio. In addition, the media included 1% Non-Essential Amino Acids (NEAA) (Gibco), 2% B27 serum (Gibco), 1% N2 (Gibco), 1% antibiotic-antimycotic (Gibco), 0.2 μM LDN193189 (Cayman Chemical), 3 μM CHIR99021 (Xcess bioscience), and 10 μM SB431542 (Cayman Chemical). Cultures were fed daily with 2 ml per well of fresh media at room temperature. On day 6 of differentiation, the resulting neuroepithelial cells were collected and re-plated at a density of 7.5e5 cells/well in a 6-well plate in Stage 2 media. Stage 2 media contained 1:1 IMDM (Gibco) and F12 (Gibco), 1% NEAA (Gibco), 2% B27 (Gibco), 1% N2 (Gibco), 1% antibiotic-antimycotic (Gibco), 0.1 μM All-trans retinoic acid (Stemgent), 1 μM Sonic Hedgehog Agonist (SAG) (Cayman Chemical), 0.2 μM LDN193189 (Cayman Chemical), and 10 μM SB431542 (Cayman Chemical). The cells were then cultured in Stage 2 iMN media for 6 days and then in Stage 3 iMN media from day 11 onward. Stage 3 media contained 1:1 IMDM (Gibco) and F12 media (Gibco), and 1% NEAA (Gibco), 2% B27 (Gibco), 1% N2 (Gibco), 1% antibiotic-antimycotic (Gibco), 0.1 μM SAG (Cayman Chemical), 0.5 μM all-trans retinoic acid (Stemgent), 0.1 μM compound E (Calbiochem), 2.5 μM DAPT (Cayman Chemical), 200 ng/ml Ascorbic Acid (Sigma), 10 ng/ml BDNF (Peprotech), 10 ng/ml GDNF (Peprotech). Cells were maintained in Stage 3 media and fed 2 ml fresh room temperature media every other day until 20 days post-differentiation. Quantitative real-time PCR of Acetylcholine transferase was used to validate that differentiation was successful.

Motor neurons were characterized by immunofluorescent staining. Ninety-five % of the differentiated neurons (with long neurites, grew out from the neurospheres) were ChAT positive, indicating the lower motor neuron identity. These cells were also ISL-1 positive, to a lesser degree. We could not identify the other cells (<5%), but based on their morphology, they appeared to be glial cells, though at this stage they were still GFAP negative. It was difficult to clarify all cell types in the center of the aggregated neutrospheres precisely as they were compact, but they were mostly undifferentiated neural stem/progenitor cells with positive nestin staining.

## Immunofluorescence in human brains

Immunofluorescence was performed on 5 μm-thick paraffin sections sourced from cortical tissues with antibodies against TDP-43, ASRGL1 and NeuN (Supplementary Table 4). Briefly, the sections were first deparaffinized using standard Xylene/Ethanol/Rehydration protocol followed by antigen unmasking with a 10-min heat mediated antigen retrieval step in 10 mM Tris/EDTA buffer pH 9.0 (Tris/EDTA buffer) using an 800 W microwave set at 100% power. The sections were then incubated with Human BD Fc Blocking solution (BD Biosciences) to saturate endogenous Fc receptors, and then in True Black Reagent (Biotium) to quench intrinsic tissue autofluorescence. The sections were then immunoreacted for 1 h at room temperature using 1–5 μg/ml cocktail mixture of primary antibodies. This step was followed by washing off excess primary antibodies in PBS supplemented with 1 mg/ml bovine serum albumin (BSA) and staining the sections using a 1 μg/ml cocktail mixture of the appropriately cross-adsorbed secondary antibodies (purchased from either Thermo Fisher, Jackson ImmunoResearch or Li-Cor Biosciences) conjugated to one of the following spectrally compatible fluorophores: Alexa Fluor 488, Alexa Fluor 546, Alexa Fluor 594, Alexa Fluor 647 or IRDye 800CW. After washing off excess secondary antibodies, sections were counterstained using 1 μg/ml 4′,6-diamidino-2-phenylindole (DAPI, Thermo Fisher Scientific) for visualization of cell nuclei. Slides were then coverslipped using Immu-Mount medium (Thermo Fisher Scientific) and imaged using a multi-channel wide field epifluorescence microscope (see below).

## Immunofluorescence image acquisition

Images were acquired from whole specimen sections using the Axio Imager.Z2 slide scanning fluorescence microscope (Zeiss) equipped with a 20X/0.8 Plan-Apochromat (Phase-2) non-immersion objective (Zeiss), a high resolution ORCA-Flash4.0 sCMOS digital camera (Hamamatsu), a 200 W X-Cite 200DC broad band lamp source (Excelitas Technologies), and 6 customized filter sets (Semrock) optimized to detect the following fluorophores: DAPI, Alexa Fluor 488, Alexa Fluor 546, Alexa Fluor 594, Alexa Fluor 647 and IRDye 800CW. Image tiles (600 × 600 μm viewing area) were individually captured at 0.325 micron/pixel spatial resolution, and the tiles seamlessly stitched into whole specimen images using the ZEN 2 image acquisition and analysis software program (Zeiss), with an appropriate color table having been applied to each image channel to either match its emission spectrum or to set a distinguishing color balance. Pseudocolored stitched images acquired from antibody staining and imaging were then exported to Adobe Photoshop and overlaid as individual layers to create multicolored merged composites.

## Deamidation analysis of TDP-43 with NGOME-Lite

Prediction of spontaneous non-enzymatic deamidation of asparagine residues and isoaspartate formation in TDP-43 was performed using the software NGOME-Lite[22]. The program uses sequence based secondary structure and considers local flexibility and presence of alpha helices for identifying asparagine residues prone to deamidation (predicted by JPred; www.compbio.dundee.ac.uk/jpred). The false positive discovery rate was set up at 5%.

## 3D modeling and calculation of protein energy

Using the Basic Local Alignment Search Tool (BLAST), the amino acid sequence for TDP-43 was aligned to available protein sequences in the protein data bank (PDB). Each PDB sequence was downloaded using the Schrodinger Suites software package (www.schrodinger.com) and the default settings in the protein preparation wizard were used to prepare the protein sequence for analysis. The Gibbs free energy of the peptides was calculated using the VSGB solvation model and OPLS3e force field and is reported in kcal/mol. Each sequence was then manually modified to convert the desired asparagine residues to an isoaspartate by deleting the backbone peptide bond, deleting the amine from the asparagine residue, creating a bond between the backbone nitrogen and the carbonyl of the asparagine residue, and converting the backbone carbonyl to a carboxylate. The energy of the entire structure was minimized, then calculated using the above specifications.

Interaction of TDP-43 and ASRGL1 was analyzed by docking with the ClusPro 2.0 web server[75].

## Detection of asparagine deamidation of TDP-43 by mass spectrometry

Immunoprecipitation of brain samples was performed prior to mass spectrometry analysis. Brain cortex samples (300 mg) were lysed and homogenized in RIPA buffer (Thermo Fisher) and incubated overnight with 5 ug of C-terminal TDP-43 antibody (TDP43 G400 Antibody, Cell Signaling #3448). Samples were washed 3 times with RIPA buffer + 1% NP-40 (Sigma-Aldrich) and immunoprecipitated with 25 µl of A/G beads (Thermo Fisher). On-beads samples (11.73 ± 6.34 µg of protein) were reduced with 10 mM Tris(2-carboxyethyl) phosphine hydrochloride (TECP), alkylated with 10 mM N-Ethylmaleimide (NEM), and digested with trypsin. Trypsin digestion was performed with a 1:20 trypsin:sample ratio at 37 °C for 8 h. The digestion was repeated twice. The incubation time for the 2nd and 3rd digestion was 18 and 8 h, respectively. Trypsin digestion was quenched by adjusting the pH to 3 with 10% formic acid in HPLC water. Tryptic peptides were extracted with 0.1% TFA in HPLC water, desalted using Waters Oasis HLB plate, injected into a nano-LC-MS/MS system where an Ultimate 3000 HPLC (Thermo-Dionex) was coupled to an Orbitrap Lumos mass spectrometer (Thermo Scientific) via an Easy-Spray ion source (Thermo Scientific). Peptides were separated on an ES802 Easy-Spray column (75-µm inner diameter, 25 cm length, 3 µm C18 beads; Thermo Scientific). The composition of mobile phases A and B was 0.1% formic acid in HPLC water, and 0.1% formic acid in HPLC acetonitrile, respectively. The mobile B amount was increased from 3% to 30% in 66 min at a flowrate of 300 nl/min. Thermo Scientific Orbitrap Lumos mass spectrometer was operated in data-dependent mode. The MS1 scans were performed in orbitrap with a resolution of 120 K at 200 $m/z$ and a mass range of 400–1500 $m/z$, and an automatic gain control (AGC) value of $2 \times 1e5$. The MS2 scans were conducted in ion trap with an AGC target of $3 \times 1e4$. The isolation width was 1.6 $m/z$, and the dynamic exclusion window was 12 s.

Xcalibur RAW files were converted to peak list files in mgf format using Mascot Distiller (version 2.7.1.0). Database search was performed using Mascot Daemon (2.6.0) against Sprot Human database. Enzyme: Trypsin. Fixed modifications: Nethylmaleimide (C). Variable modifications: Acetyl (Protein N-term), Deamidated (NQ), Deamidated (Protein N-term F), Deamidated (R), Oxidation (M). Mass values: Monoisotopic. Peptide mass tolerance: ± 10 ppm. Fragment mass tolerance: ± 0.6 Da. Max missed cleavages: 4.

## Proximity ligation assay (PLA)

Proximity ligation assay (PLA) to detect ASRGL1 and TDP-43 interaction was performed with the Duolink Proximity Ligation Assay kit (Millipore Sigma) in HEK293 transfected cells and brain slides, following manufacturer's instructions. Briefly, samples were blocked with Duolink® Blocking Solution for 60 min at 37 °C. Then, primary antibodies were applied (ASRGL1: Atlas HPA05572 (rabbit). TDP-43: Proteintech (mouse)) at 1:200 in Duolink antibody diluent, and samples were incubated overnight in a humidity chamber at room temperature. Then, minus and plus probes were diluted 1:5 in the Duolink antibody diluent. The slides were washed twice in wash buffer A and incubated with the probes for 1 h at 37 °C. Samples were again washed twice in wash buffer A and incubated with ligase for 30 min at 37 °C. Samples were once again washed twice in wash buffer A. Then, polymerase was added, and slides were incubated for 100 min at 37 °C. Slides were washed twice in wash buffer B 1x and once in wash buffer B 0.01x. Slides were mounted with Duolink mounting media with DAPI and analyzed by confocal microscopy. Since the levels of ASRGL1 are higher in normal controls than in ALS individuals, negative controls were produced from slides from normal controls: (1) No primary antibodies, (2) Only TDP-43 antibody, (3) Only ASRGL1 antibody, (4) TDP-43 antibody + Control isotype IgG.

## Determination of binding interactions between ASRGL-1 and TDP-43 with differential scanning fluorimetry

Differential scanning fluorimetry (DSF) or protein thermal shift assays[76] were completed by use of an Applied Biosystems QuantStudio 6 Flex real time PCR instrument with 384-well plate or a QuantaBio qPCR instrument. In these reactions, the assay plates contained a final volume of 20 µl with PBS (10 mM phosphate and 150 mM NaCl [pH 7.4]), Protein Thermal Shift Dye kit (Thermo Fisher Scientific; cat# 4461146, diluted 1:125 in kit diluent) and protein samples: full-length ASRGL1expressed in E coli and purified (>90%) by chromatography (Abcam; ab187483), native form full-length TDP-43 expressed in E coli and purified (>95%) by chromatography (R&D Systems; AP190100) and, as negative control, full-length TDP-43 denatured through chaotropic conditions (urea), expressed in E coli and purified by chromatography (Abcam ab156345). The plate was sealed and mixed briefly, followed by plate centrifugation at 1000 rpm for 2 min. After 30-min incubation at 25 °C, the plate was then subjected to thermal shift by ramping the temperature from 25 °C to 99 °C at 0.05 °C increments per second. The relative fluorescence emitted by the thermal shift dye was recorded during the temperature ramp phase and plotted versus temperature. The derivative of these thermal melt curves was determined, and Tm calculated from these data. The Tm values were plotted as a function of TDP43 concentration in GraphPad Prism, and the inflection point of the curve determines the binding affinity of ligand to binding protein. The experiments were completed with $n = 4$ replicate samples per treatment. Each experiment was repeated 3 times.

## DNA constructs

ASRGL1 shRNAs (sequences in Supplementary Table 5) for the human and the mouse genes were cloned in pcDNA3.1 vectors (Origene). A lentiviral construct encoding 4 shRNAs (Supplementary Table 5) against the mouse ASRGL1 gene was generated and packed into AAV9 viral particles (Vigene) for ASRGL1 silencing in mouse cells by viral transduction (construct map in Supplementary Fig. 11). A lentiviral construct encoding 4 scrambled shRNAs and packed in AAV9 viral particles was used as control (Vigene). A Myc-tagged intrabody against misfolded TDP-43[24] was cloned in a pcDNA 3.1 vector (Supplementary Fig. 12a, b) and validated by the detection of ALS-associated TDP-43 mutants prone to misfolding (M337V and Q331K)[77] (Supplementary Fig. 12c, d). The HERV-K (HML-2) whole genome consensus sequence[78] was synthesized and cloned into pcDNA3.1 vector (Invitrogen). A lentiviral construct encoding 4 HERV-K (HML-2) LTR–targeting gRNAs (G8: gatagggaaaaaccgcctta; G10: aaagcagtattgctgcccgc; G15: agtagatggagcatacaatc; G42: Tcctgcctgtccctgggcaa) and nuclease-null Cas9 linked to the transcription activator domain VP160 was generated (CRISPR/dCAS9-VP160). The dCas9-NLS-3×HA-VP6160 vector (Addgene) served as the backbone.

## Neurons and neural stem cultures

IPSCs-derived neural stem cells modified to express tdTomato protein[18] and unmodified NSCs were maintained in proliferation media: KnockOut DMEM/F-12, Glutamax 2 mM, StemPro Neural Supplement 2% (Thermo Fisher), bFGF (20 ng/ml) and EGF (20 ng/ml) (Peprotech); or differentiated into neurons with differentiation media: DMEM/F-12, Glutamax 2 mM (Thermo Fisher), bovine serum albumin 1.8% (Thermo Fisher), hESC supplement (2%) (Thermo Fisher), BDNF (20 ng/ml) and GDNF (20 ng/ml) (Peprotech).

## Transfections

Neural cells were transfected by nucleofection with a 4D unit (Lonza) and Amaxa P3 or P4 kits. DC100 and DN100 programs were used to transfect neural stem cells and neurons, respectively as per manufacturer's protocols. HEK293 were transfected with Lipofectamine 3000 (Thermo Fisher), following manufacturer's instructions. One µg

of DNA per 2 million cells was used for TDP-43, TDP-43 intrabody and ASRGL1 plasmids. For the shRNAs constructs, we used 0.66 μg of DNA of each plasmid per 2 million cells. HML-2 plasmid was used at 0.5, 1 and 2 ug per 2 million cells.

## Neurotoxicity assays

Human neurons derived from an IPSc line modified to expressed tdTomato fluorescent protein[18] and mouse neurons derived from neural stem cells modified to express GFP (Catalogue number: MUBNF-01101. As One International. Santa Clara. California) were transfected at 7 days of differentiation using the P4 nucleofector kit (Lonza) and DN100 program according to manufacturer's instructions. Combinations of 3 different shRNAs against *ASRGL1* were used (0.3 μg of DNA per plasmid/million cells). Control scrambled shRNAs plasmids were always used at the maximum concentration of the sum of the experimental plasmids to account for the toxicity produced by the DNA itself. HML-2 plasmid was used at 0.5 μg of DNA per million cells. After transfection, neurons were plated in 96-well plates and 24 h later, labeled cells were automatically counted in a Cytation 5 imager. For the transduction experiment, AAV-9 viral particles packing shRNAs against *ASRGL1* or scrambled shRNAs were added to neuronal cultures at 3 different multiplicity of infections (MOI) (1000, 5000 and 10,000 MOI). Twenty-four hours later, cells were imaged with a fluorescence microscope (Evos, Thermo Fisher). For the experiment of transfection with TDP-43 peptides and ASRGL1 plasmid, neurons were transfected with Nucleofector (Lonza) and plated in 96 wells plates. 24 h later images were taken by confocal microscopy. We acquired 4 fields per well and 6 wells per treatment for each experiment. They were captured at ×10 magnification in a preset 2 × 2 array in the well (same settings for every well). The mean neurite length was determined from these images using Molecular Devices MetaXpress Neurite Outgrowth software. All experiments were repeated at least 3 times.

## Western blot analysis

NSCs, neurons (1 million cells) or brain cortex samples (50 μg) were lysed with N-PER buffer (Thermo Fisher) following manufacturer's recommendations. Protein concentration in the lysates was measured with the Pierce BCA protein assay kit and all samples to be compared were adjusted to the same concentration and mixed with SDS- 10% beta mercaptoethanol. Samples were denatured at 95 °C for 5 min at 900 rpm in a thermo shaker. They were then resolved in a 4–12% bis-tris gel (Invitrogen) at 200 V for 45 min and then transferred to a PDVF membrane (iBlot PDVF stack; Thermo Fisher) using an iBlot blotting system (Thermo Fisher) for 7 min. Then, membranes were blocked with PBS-5% skimmed milk for 1 h and incubated at 4 °C with the primary antibodies (1:1000 in TBS-5% albumin-0.02% sodium azide) overnight. The next day blots were washed 3 times in PBS-0.05% Tween-20 for 5 min and incubated with secondary HPR-conjugated antibodies (Cell signaling) for 1 h (1:2500 in PBS-0.05% Tween-1% Skimmed milk) in PBS. Blots were washed again 3 times and chemiluminescence was developed with Supersignal West Dura extended duration signal kit (Thermo Fisher) and imaged with Protein Simple imager, except for misfolded, phospho-TDP-43 and TDP35 blots, which were developed with Supersignal West Femto (Thermo Fisher). Images were processed for quantification of chemiluminescence with ImageJ software. Antibodies used in Supplementary Table 4. Protein expression was calculated as the ratio between the intensity of the band of the target protein and the intensity of the band of the loading control protein. The percentage of the ratio in relation to the control was calculated for every treatment in an experiment.

## Cloning of an intrabody against misfolded TDP-43

The Gibson Assembly cloning kit (New England Biolabs) was used to insert a previously described Myc-tagged intrabody sequence[24] (IDT) into a pcDNA 3.1(+) vector (Thermo Fisher) at NheI (Supplementary

Fig. 12a, b). The Gibson Assembly® Chemical Transformation Protocol (E5510, NEB) was then used to transform and plate bacteria. Colonies were selected and propagated overnight at 37 °C in LB broth containing 1 ng/ml of ampicillin. Plasmid DNA was isolated using the Plasmid Plus Midi Kit (Qiagen). Restriction digest with BamHI/NdeI and sequencing of the plasmid with a T7 promoter primer was used to verify the insert in the plasmid. Binding of the intrabody to misfolded TDP-43 was validated transfecting HEK293 with either wild-type TDP-43 or two ALS-associated TDP-43 mutants prone to misfolding (M337V) and (Q331K)[77], and by measuring the levels of immunoprecipitated TDP-43 by western blotting (Supplementary Fig. 12c, d).

## Co-transfection, immunoprecipitation and western blot analysis to detect misfolded TDP-43

iPSC-derived neurons were co-transfected with (1) a construct encoding an intrabody against misfolded TDP-43 (1 μg of DNA per million cells), (2) plasmids encoding shRNAs against *ASRGL1* (sequences B, C and D; Supplementary Table 5) or scrambled shRNAs (total of 1 μg of DNA per million cells) and (3) a plasmid encoding full-length TDP-43 (1 μg of DNA per million cells). After 24 h, cells were lysed in N-PER buffer (Thermo Fisher) containing protease inhibitor cocktail (Roche). Pierce BCA Protein Assay Kit (Thermo Fisher) was used to quantify protein concentration in the lysates. Concentration was adjusted to have the same amount of protein in all lysates (15 ug). Lysates were incubated with Pierce™ Anti-Myc magnetic beads (25 μl) (Thermo Fisher) at 4 °C overnight. After incubation, the beads were recovered with a magnet and washed 3 times with N-PER buffer. Immunoprecipitates were boiled at 95 °C with shaking in 2x LDS with 10% beta-mercaptoethanol. The denatured products were separated on NuPAGE 4–12% Bis-Tris Gel (Invitrogen) and proteins were transferred onto a PVDF membrane using iBlot® 2 PVDF Mini Stack (Invitrogen). The membrane was blocked in 5% fat-free milk and then incubated with anti-Myc and anti-TDP43 antibodies and developed with HRP-secondary antibodies. The blots were visualized with SuperSignal™ West Femto Maximum Sensitivity Substrate (Thermo Fisher). ImageJ software was used for densitometry analyses.

## Immunocytochemistry

Cells were fixed with 4% paraformaldehyde for 15 min, washed twice with PBS pH 7.4 for 5 min and incubated with blocking solution (10% normal donkey serum in 1X PBS with 0.3% Triton) for 1 h at room temperature. Cells were incubated with the primary antibodies TDP-43, ASRGL1 and NeuN (Supplementary Table 4) at 1:500 dilution in PBS with 1% donkey serum at 4 °C overnight. Then, cells were washed 3 times in PBS for 5 min each and incubated with the secondary antibodies (Thermo Fisher) at 1:500 in PBS with 1% donkey serum for an hour. After 3 washes with PBS, cells were mounted with Fluorogel II with DAPI mounting media (Electron microscopy Sciences). Slides were imaged using an LSM 510 META laser scanning confocal microscope (Carl Zeiss). Cells incubated with secondary antibody only were used as controls.

## Transfection with TDP-43 peptides

C-terminal TDP-43 peptides were synthesized by Creative Peptides (NJ, USA). Sequences were as follows: TDP-43 wild-type: SYSGSNS-GAAIGWGSASNAGSGSGFNGGFGSSMDSKSSGWGM; Iso-aspartyl-modified: [β-Asp]SYSGSNSGAAIGWGSASNAGSGSGFNGGFGSSMDSK SSGWGM. One million of tdTomato-expressing neurons per cuvette were transfected with TDP-43 peptides (5 mM) using Amaxa Nucleofector P3 kit (Lonza) with the DN100 program. For the rescue experiment, neurons were co-transfected with an ASRGL1 plasmid (1 μg per million cells) and the TDP-43 peptides. Number of neurons and neurite length was automatically analyzed by confocal microscopy. We acquired 4 fields per well and 6 wells per treatment for each experiment. They were captured at ×10 magnification in a preset 2 × 2

array in the well (same settings for every well). The mean neurite length was determined from these images using Molecular Devices MetaXpress Neurite Outgrowth software. All experiments were repeated at least at least 3 times.

## Adeno-associated virus 9 (AAV9) viral particles

AAV9 viral particles packing a construct encoding four different shRNAs against the murine sequence of *Asrgl1* (Supplementary Table 5) or four scrambled shRNAs were prepared by Vigene (Rockville, Maryland. USA) and provided at $10^{14}$ particles/ml (vector map in Supplementary Fig. 11).

## Animal study

The study was approved by the NINDS/NICD/NCCIH Institutional Animal Care and Use Committee (Protocol number: 1331-20).

## Stereotaxic injections

Three-month-old adult female C57BL/6 mice were purchased from Jackson Laboratory (Bar Harbor, Maine) and maintained in our breeding colony. All experiments involving mice were performed according to the recommendations in the "Guide for the Care and Use of Laboratory Animals of the NIH". Mice were housed in a pathogen-free barrier facility with a 12-h light/12-h dark cycle and ad libitum access to food and water. Since the study was conducted with a limited number of animals ($n = 5$ in each group), only females were used to eliminate the potential sex impact. For the stereotaxic intracerebral injection, mouse anesthesia was induced with isoflurane (3–4%) via the inhalational route in plastic induction chamber, and then maintained during the placement of the intracranial injection with the use of 1–3% isoflurane administered via face mask. Once proper anesthesia was established with the animal lacking a righting response, animals were placed in a stereotaxic apparatus. A midline skin incision was made over the skull followed by the opening of a small burr hole in the skull over the injection site. The target injection site was layer 5 of the motor cortex of the right hemisphere. The coordinates for the right motor cortex were: 1.5 mm lateral, 1.1 mm anterior, 1.6 mm ventral, based on the mouse brain atlas of Paxinos and Watson, 2012. Animals were randomly divided into two groups, to receive AAV9 viral particles packing ASRGL1-shRNAs or scrambled shRNAs. Two µl of injection solution ($10^{13}$ AAV9 viral particles/ml) was injected at a rate of 0.1 µl/min using a micro-injector. The needle (10 µl volume syringes, Hamilton) was left to sit for 3 min after injection to allow for diffusion of injection solution. The needle was then slowly withdrawn to minimize backflow. The incision was closed with Vetbond. We allowed the animals to recover on a warming pad before placing it back in its housing cage. The mice were then monitored closely for self-trauma of the incision. Analgesia was provided after surgery; Bupivacaine (0.5%, 1–2 drops to incision site, locally) and Buprenex (0.05–0.1 mg/kg, subcutaneously, every 8–12 h for 24–48 h).

## Preparation of mouse brains

Mice were deeply anesthetized with ketamine (100 mg/kg) and xylazine (10 mg/kg) intraperitoneally and perfused transcardially with 1× PBS, which was followed by ice-cold fixative (4% paraformaldehyde–0.1 M phosphate buffer, pH, 7.4). Brains were dissected and post-fixed overnight at 4 °C. Fixed tissues were cryoprotected in 10%, 20% and 30% sucrose at 4 °C; every incubation was 24 h. Tissues were then frozen until ready for sectioning. Coronal serial sections were cut to a thickness of 40 µm.

## Immunohistochemistry of mouse brains

For immunofluorescent staining, mouse brain sections were washed in 1x tris-buffered saline (TBS) prior to incubation in blocking solution (TBS with 0.5% Triton-X and 2.5% donkey serum). Primary antibodies anti-Ctip2, NeuN, ASRGL1 and TDP-43 (Supplementary Table 4) were diluted in blocking solution and incubated at 4 °C. After washing with 1xTBS with 0.05% Triton-X, sections were incubated with fluorescent conjugated secondary antibodies (1:250, Jackson ImmunoResearch), followed by washing and counterstaining with DAPI to label all nuclei. Stained sections (6 per mouse) were mounted on slides and viewed with a Zeiss confocal laser microscope. All slides were analyzed by two independent blinded investigators.

## Quantitative PCR of ASRGL1, HML-2 env, and UNC13A cryptic exon expression

Total RNA was extracted from motor neuronal lysates or brain lysates with Rneasy Mini kit (Qiagen), following manufacturer's recommendations. The RNA was treated with Turbo DNA-free kit (Thermo Fisher) following manufacturer's recommendations to eliminate remaining DNA. RNA concentration was measured by Nanodrop and all samples were adjusted to the same concentration. Reverse-transcription was performed with the SuperScript™ III First-Strand Synthesis System (Thermo Fisher), adding a no-RT control to account for DNA contamination. Quantitative PCR (qPCR) was performed in a Fast qPCR system, VIA VII (Thermo Fisher). *ASRGL1* was analyzed with the following reaction in a final volume of 20 µl: 7 µl of PCR water, 10 µl of TaqMan™ Fast Advanced Master Mix (Thermo Fisher), 1 µl of a set of primers and probe specific for *ASRGL1* (Hs05025729_m1; Invitrogen) and 2 µl of cDNA. *18s*RNA was analyzed as housekeeping gene (Thermo Fisher). *HML-2 env* was analyzed with the following reaction in a final volume of 20 µl: 6 µl of PCR water, 10 µl of Fast Sybr Green Master Mix (Thermo Fisher), 500 nM primers and 2 µl of cDNA. *HPRT1* and *RPLPO* were analyzed as housekeeping genes. Primer sequences: *HML-2 env* forward: 5′ CTGAGGCAATTGCAGGAGTT 3′; *HML-2 env* reverse: 5′ GCTGTCTCTTCGGAGCTGTT 3′; *UNC13A-CE* forward: 5′TGG ATG GAGAGA TGGAACCT-3′; *UNC13A-CE* reverse: GGGCTGTCTCAT CGTAGTAAA; *HPRT1* forward: 5′ GCTATAAATTCTTTGCTGACCTGCTG 3′; *HPRT1* reverse: 5′ AATTACTTTTATGTCCCCTGTTGACTGG 3′; *RPLPO* forward: 5′ TCTACAACCCTGAAG TGCTTGAT 3′; *RPLPO* reverse: 5′ CAATCTGCAGACAGACACTGG.

The data was analyzed with the Via7 software. RNA expression was calculated with the Delta Delta Ct method (Livak method[79]). "Fold change" in all graphs presenting RNA expression is calculated in relation to the average of the controls[79].

## Statistics and reproducibility

No statistical method was used to predetermine sample size. No data were excluded from the analyses. The experiments were not randomized. Where indicated the investigators were blinded to outcome assessment. Data was analyzed with GraphPad Prism version 6 (GraphPad). The Kolmogorov–Smirnov test of normality was applied to all data sets to determine whether a data set was normally distributed. The *F* test or Bartlett's test was used to determine equality of variances between groups. When groups demonstrate a normal distribution and were homoscedastic, differences between means were assessed by unpaired two-sided Student's *t* test for two groups or One-way ANOVA followed by post hoc testing with Bonferroni's correction for multiple comparison for three or more groups. For comparison of three or more groups that did not have equal variances, the Brown-Forsythe ANOVA test followed by post hoc testing with Bonferroni's correction for multiple comparison was used[80]. For comparison of two groups that did not have equal variances the Mann–Whitney test was used. All pairwise comparisons were performed with two-sided tests. Values in graphs are expressed as mean ± standard error mean (SEM) for parametric comparisons and as median with interquartile range (IQR) for non-parametric tests. Comparison of frequencies was performed by Chi-square test. Correlation of two variables was assessed by the Pearson's *R* test. For the analysis of TDP-43 and ASRGL1 binding by thermal shift a non-linear regression (sigmoidal) was performed. Statistical significance was achieved when *p* value < 0.05.

**Reporting summary**

Further information on research design is available in the Nature Portfolio Reporting Summary linked to this article.

## Data availability

The datasets generated during current study have been deposited in the Figshare database under accession "TDP-43 proteinopathy in ALS is triggered by loss of ASRGL1 and associated with HML-2 expression". Public databases have been used for sequence alignment of proteins (PDB, Sprot Human database, Uniprot) and nucleic acids (GRCh38), protein interactions (BioGRID), and gene ontology (Gene Ontology (GO) Consortium). RNA sequencing data were obtained from the ALS Consortium of the New York Genome Center and would available upon request: https://www.nygenome.org/contact/. Source data are provided with this paper.

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

## Acknowledgements

We are grateful to the NIH NeuroBiobank and the institutions which donated brain samples for this study: University of Maryland Brain and Tissue Bank, Harvard Brain Tissue Resource Center, Brain Tissue

Donation Program at the University of Pittsburgh, University of Miami Brain Endowment Bank, Human Brain and Spinal Fluid Resource Center, Mount Sinai Neuropathology Brain Bank, Sepulveda Research Corporation, and the Department of Veteran's Affairs Biorepository Brain Bank (VA Merit Review BX002466). We also thank the ALS Consortium of the New York Genome Center for providing the RNA sequencing data used for the study and the Massachusetts Institute of Technology. The work at NYGC is supported by the ALS Association (ALSA) and The Tow Foundation. This research was supported (in part) by the Division of Intramural Research of the NIH, NINDS. The content is solely the responsibility of the author(s) and does not necessarily represent the official views of the National Institutes of Health. We also acknowledge funding from the ALS Association (Grant 20-SI-559, A.N. and M.G.M.), Agencia Nacional de Promoción Científica y Tecnológica (PICT 2015-1213 grant to I.E.S.) and Consejo Nacional de Investigaciones Científicas y Técnicas (I.E.S. is a CONICET career investigator, J.R.L. is a CONICET career technical staffer).

## Author contributions

M.G.M. and A.N. planned and oversaw all aspects of the study. M.G.M., S.F. and C.R. performed and analyzed most of the experiments. E.R.S. and A.M. analyzed microscopy images. T.D. differentiated the motor neurons from IPSc lines and contributed to the analysis of RNAseq data. Y.H.H.C. performed the experiments with the intrabody against misfolded TDP-43. R.A. performed the in silico analysis of the isoaspartates substitutions in TDP-43. N.P. performed most of the RNAseq analysis. N.M. and M.B. ran some of the neurotoxicity assays. B.D. assisted in various experiments involving transfection and western blotting. D.M. performed some of the immunofluorescence staining. M.-H.L. performed the DAB staining. H.W. carried out the stereotaxic injections in mice. U.S. assisted in obtaining and organizing human samples and clinical data. W.L. and K.S. designed and cloned some of the plasmids. J.R.L. and I.S. designed and ran the NGOME in silico assay. Y.L. performed the mass spectrometry analysis. L.A.S. provided substantial contribution to the interpretation of the data. S.P., M.K. and M.H. provided the single-cell RNAseq data and performed the analysis. J.S. performed neurotoxicity assays, captured, and analyzed confocal microscopy images, performed and analyzed the thermal shift assays and contributed substantially to the interpretation of the data. M.G.M. and A.N. wrote the manuscript with input and substantial revisions from all authors.

## Competing interests

The authors declare no competing interests.

## Additional information

[1]Section of Infections of the Nervous System, National Institute of Neurological Disorders and Stroke, National Institutes of Health (NIH), Bethesda, MD, USA. [2]Struttura Complessa Microbiologia e Virologia, Azienda Ospedaliera Universitaria Sassari, Sassari, Italy. [3]Translational Neuroscience Unit, National Institute of Neurological Disorders and Stroke, National Institutes of Health (NIH), Bethesda, MD, USA. [4]Flow and Imaging Cytometry Core Facility, National Institute of Neurological Disorders and Stroke, National Institutes of Health (NIH), Bethesda, MD, USA. [5]Neuro-Oncology Branch, National Cancer Institute (NIH), Bethesda, MD, USA. [6]Centro de Investigación Veterinaria de Tandil (CIVETAN), CONICET-CICPBA-UNCPBA, Facultad de Ciencias Veterinarias, Universidad Nacional del Centro (FCV-UNCPBA), Tandil, Argentina. [7]Protein Physiology Laboratory, Departamento de Química Biológica, Facultad de Ciencias Exactas y Naturales and IQUIBICEN-CONICET, Universidad de Buenos Aires, Buenos Aires, Argentina. [8] Centre de Biologie Structurale, Centre national de la recherche scientifique (CNRS), Montpellier, France. [9]Protein/Peptide Sequencing Facility, National Institute of Neurological Disorders and Stroke, National Institutes of Health (NIH), Bethesda, MD, USA. [10]Massachusetts Institute of Technology, Cambridge, MA, USA. ✉e-mail: natha@ninds.nih.gov

