## [Peer Review File · Nature Communications]

TDP-43 proteinopathy in ALS is triggered by loss of ASRGL1 and associated with HML-2 expression.Reviewer #1 (Remarks to the Author):

This is an interesting paper from the group of Avindra Nath. ASRGL1 levels had been previously described to inversely associate to those of the HML-2 endogenous retrovirus, that is encoded within an ASRGL1 intron on the antisense strand. This was not a common finding amongst genes with HML-2, and is now followed upon in the present paper.

In this work, the consequences of ASRGL1 loss in ALS are investigated and linked with TDP-43 pathology. Specifically, ASRGL1 is found to be decreased in ALS CNS; TDP-43 and ASRGL1 interact, and in absence of the latter TDP-43 degradation is impaired leading to the typical cytoplasmic accumulations.

A number of important points need to be addressed.

- ASRGL1 reduction in ALS. Authors should include measurements for other neurodegenerative conditions. Usually the main driver of expression changes in ALV vs control post mortem brains is cell type composition changes. Is the reduction specific to ALS and associated with TDP-43 pathology? How is ASRGL1 in a condition as PSP, with neuronal loss, but no TDP-43 pathology?
- Also, how is expression in other unaffected regions, such as the cerebellum?
- The staining of patient and control brains in figure 1c is not fully convincing. ALS cases appear to have a generally lower DAB staining.
- Figure 1h-l. It appears that nearly 100% of neurons and glia have TDP-43 mislocalization. This is not in line with previously reported, and TDP-43 pathology needs to usually be carefully searched to be found. This contradiction with literature and pathology practice needs to be resolved.
- In Figure 2i, it appears that levels of interaction between TDP-43 and ASRGL1 in ALS are similar to background. Have technical controls been always performed?
- It would be great to see the data from the mutant iPSC. These have been very rarely reported to have TDP-43 pathology, which should instead occur if ASRGL1 is reduced.
- Does AZT (and also re-expression of ASRGL1) impact on TDP-43 mislocalisation?
- Correlation between ASRGL1 and HML-2 does not appear to be meaningful (Figure 5i).

Reviewer #2 (Remarks to the Author):

In this manuscript the authors investigate whether the loss of asparaginase-like 1 protein (ASRGL1) contribute to the development of ALS by directly affecting TDP-43 protein misfolding, impaired degradation and consequently cytoplasmic aggregation. Furthermore, HERV-K (HML-2) overexpression (also previously reported in ALS by the same group) is suggested to induce such ASRGL1 silencing. The manuscript proposes a model that encompasses and links together diverse findings/mechanisms to reach this conclusion: 1) ASRGL1 is decreased in the cortex of ALS patients which is accompanied by cytosolic aggregation of TDP-43; 2) the loss of ASRGL1 is responsible for increased isoaspartate formation of TDP-43 and, when present, its interaction with TDP-43 in the perinuclear region promotes protein degradation and protein aggregation in the cytoplasm is prevented; 3) silencing of ASRGL1 with shRNA or by activating HML-2 leads to neuron death in mice and neuronal cell cultures.

The manuscript is well written and easy to understand and the hypothesis put forward by the authors are interesting and novel. However, the presented data is overall not fully convincing and there are several control experiments are missing. Overall, the manuscript appears to preliminary in its current form to warrant publication.

Specific comments

1. The RNA-seq analysis in Fig 1 is convincingly showing that ASRGL1 is downregulated in ALS tissue.

However, single-cell analysis (using for example available snRNA-seq data from ALS tissue) would be a very valuable addition to clarify in which cells ASRGL1 is expressed in control individual and in what cell-types ASRGL1 expression is reduced in ALS.

2. The IHC pictures are not very convincing. For example, how come there seems to be a complete absence of staining in ALS samples, while the western blot in (Fig 1 d, e) show a difference but far from being that drastic decrease.

3. The use of iPSC-derived motor neurons is an excellent idea. However, a more thorough characterization of this iPSC derived motor neurons cultures would be helpful for the interpretation of the data. Are they lower motor neurons (referenced protocol says 30%), what are the other cell types present? This should be evaluated and discussed in more detail.

4. The authors rely heavily on quantification of cell viability. However, in order to estimate cell viability there are several methods available which all have their pros and cons. Since this experimental output is essential in the manuscript the measure by fluorescence microscopy is not convincing. In addition, it is not clear to the reader what exact measurement is acquired with fluorescence microscopy. Please clarify, Cell count? Neurite length? Could the authors think of an additional approach to address this? For example, Flow cytometry or simple IP staining...? It is also confusing that Fig. 3s and t have a different nomenclature if the measurements for viability are the same as for the other panels in the manuscript addressing cell viability.

5. The results from the shRNA-experiments are intriguing. However, a lot of detail is missing. How were the shRNA-constructs validated. Rescue-experiments to verify a lack of off-target effects are also missing.

6. The authors mention two ways of isoaspartate formation but only address asparagine deamidation experimentally, why?

7. In fig. 2i low values for PLA are shown, how does the authors know that these values are related to the proximity between the protein and not just a reflection of the lower ASRGL1 expression, something already addressed in figure 1. If the authors want to point towards a proximity the samples are supposedly normalized, however this is not clear.

8. Why is there not a control for ASGL1+IgG in Fig.2 f and i?

9. In previous research the same authors report that "expression of HERV-K was regulated by TDP-43" Li et al. 2015. This aspect is not addressed in this manuscript, however this need to be properly discussed in relation to the suggested activation of HERV-K by ASRGL1. Are both factors involved in the HERV-K expression, etc...?

10. It is important that the authors provide more information about other factors affected by the HML-2 activation in Fig. 5, essentially but not exclusively proteins related to the HERV-K family and TDP-43. This should be examined in more detail as a general effect cannot be ruled without this data.

11. It is hard to follow what models (cell type, human brain tissue etc.) are used in which panels in most figures. Please clarify.

12. The discussion is hard to follow and disperse and would need editing.

13. On line 246 the authors claim that "TDP-43 levels decreased in the neurons treated with scrambled shRNA" which is an over interpretation of the data.

14. In Fig. 3 there is a difference in the protein levels in the TDP-43 western blots comparing 3e,g

with 3o at 0h. This needs to be explained since a variation in total TDP-43 expression would affect the phosphorylated TDP-43 results.

15. Fig 3: In several panels it is vague what % corresponds to

16. The n=values are hard to understand

17. In Fig. 2f the dots are red for ALS. Should they be black?

18. On line 191 the figure is mis-referenced. Should it be Fig 2g?

19. The reference numbers should be in superscript on line: 215, 225, 241, 279, 302, 310, 318, 362, 420, 447

20. Reference 30 needs editing

21. On line 1213 the 3 is missing in TDP-43

Reviewer #3 (Remarks to the Author):

In this study, Garcia-Montojo and colleagues identify loss of asparaginase-like 1 (ASRGL1) as a potential explanation for TDP43 pathology and neurodegeneration in ALS. The same group previously demonstrated activation of endogenous retroviruses such as HML2 in ALS samples. As shown in this manuscript, because the gene encoding ASRGL1 contains an HML2 copy, reactivation of virus leads to ASRGL1 downregulation at the RNA and protein levels. The authors also show that TDP43 is an ASRGL1 substrate. In the absence of this enzyme TDP43 becomes misfiled, accumulates in the cytoplasm and is phosphorylated, similar to what is found in ALS patients. Loss of ASRGL1 is also sufficient to reproduce neurodegeneration in cultured cells, and ASRGL1 overexpression protects against toxicity arising from TDP43 accumulation.

The authors provide a compelling argument for the connection between ASRGL1, TDP43 and neurodegeneration in ALS though a combination of *in silico*, *in vitro*, *in vivo*, post mortem, and cellular models. In general this is an exciting finding that could provide an explanation for TDP43 pathology in the setting of retrovirus activation. Even so, there are some concerns that could strengthen the findings if addressed.

Major concerns:

More information is required surrounding the detection of TDO43 asparagine deamination in ALS brains by mass spectroscopy. Is there a relationship between deamination and sex, since the proportion of males is higher in controls vs. ALS samples? How commonly were specific residues deamidated in controls vs. ALS patients? Are the effects region specific? How does this compare to other proteins, i.e. is the effect specific to TDP43?

Fig. 2i -- this is interesting, and implies that TDP43 interacts with ASRGL1 in human brain. It would be ideal if this interaction could be confirmed by a separate method (i.e. Co-IP), although this reviewer understands that such assays can be problematic.

Additionally, the results could be confounded by the reduction in ASRGL1 protein shown in Fig. 1. Is the PLA signal region dependent, meaning higher in affected regions of the CNS vs. unaffected?

Fig. 3e-j -- the authors are encouraged to show efficiency of ASRGL1 knockdown in transfected cultures. It is surprising that they were able to see such robust results by WB after transient transfection, which is not always so efficient in these cultures.

Fig. 3l -- The authors are encouraged to measure signs of TDP43 dysfunction as indicated by cryptic exon inclusion in STMN2 or UNC13A, in human neurons upon ASRGL1 knockdown.

Fig. 3o,p -- the results showing dramatic accumulation of TDP43 after only 6h are surprising. This is significantly more than what is seen with impairment of the proteasome by MG132 (which appears to have little or no effect compared to Sc shRNA). How do the authors account for this?

Fig. 3s,t -- how do the authors explain toxicity from TDP43 peptides? This is an unconventional way to introduce TDP43 toxicity, and its relevance is unclear.

Fig. 4e -- TDP43 mislocalization in ASRGL1 shRNA-treated animals is likewise impressive. Additional evidence of TDP43 dysfunction as judged by missplicing could strengthen this result (see for example PMID 29764981)

Fig. 5k, l -- these plots show that in some patients with ALS (but not all or even a majority) HML2 RNA levels are higher than controls. It is difficult to conclude that HML2 increase is a common driver of neurodegeneration in ALS with this degree of variability.

The other challenging aspect of this plot is that if only ALS patients are considered (pink dots), there is no real relationship between HML2 RNA and ASRGL1 protein levels. This is inconsistent with the authors' hypothesis.

Minor:

TDP43 phosphorylation likely has minimal effects on TDP43 trafficking or function, and if anything counteract aggregation (see PMID 35112738).

Fig. 1g and 1h -- please describe the origin of cells (i.e. sALS) within the text or figure legend.

Fig. 3a-d -- the efficacy of ASRGL1 knockdown in each case and combination should be confirmed.

Fig. 3l -- knockdown of ASRGL1 does not appear to be very efficient in the indicated cells with TDP43 mislocalization.

Sup. Fig. 3 -- why do neurons have cytoplasmic TDP43? Have they been stressed in some way, or is this a dead/dying cell?

Fig. 5k, l -- the y-axes on these plots are not intuitive. For 5k, what is the 'HML2 RNA fold change' measured in relation to? And for 5l, the legend and text describe ASRGL1 protein levels, but the plot shows 'ASRGL1 ratio'.

In the discussion, the authors stated that 'ASRGL1 rescues the neurotoxicity caused by TDP-43 C-terminal fragments' (line 417-418). This is not entirely accurate, since ASRGL1 expression rescued toxicity from a small peptide that corresponds to part of the TDP43 C-terminus. It remains unclear whether ASRGL1 can prevent toxicity from full length TDP-43 or CTFs.

TDP-43 proteinopathy in Amyotrophic Lateral Sclerosis is triggered by loss of asparaginase like 1 protein which is regulated by an endogenous retrovirus.

Responses to the reviewers' comments.

We would like to thank the reviewers for their insightful comments on our manuscript. We have addressed all their comments in a point-by-point manner. By including their suggestions, we think our study is much more robust and compelling. Please, see below our responses to their comments.

REVIEWER COMMENTS

Reviewer #1 (Remarks to the Author):

This is an interesting paper from the group of Avindra Nath. ASRGL1 levels had been previously described to inversely associate to those of the HML-2 endogenous retrovirus, that is encoded within an ASRGL1 intron on the antisense strand. This was not a common finding amongst genes with HML-2 and is now followed upon in the present paper.

In this work, the consequences of ASRGL1 loss in ALS are investigated and linked with TDP-43 pathology. Specifically, ASRGL1 is found to be decreased in ALS CNS; TDP-43 and ASRGL1 interact, and in absence of the latter TDP-43 degradation is impaired leading to the typical cytoplasmic accumulations.

A number of important points need to be addressed.

1. ASRGL1 reduction in ALS. Authors should include measurements for other neurodegenerative conditions. Usually, the main driver of expression changes in ALS vs control postmortem brains is cell type composition changes. Is the reduction specific to ALS and associated with TDP-43 pathology? How is ASRGL1 in a condition as PSP, with neuronal loss, but no TDP-43 pathology?

- **Response:** Thank you for this suggestion. To address this comment and the concern about the different DAB background intensity expressed in question number 3 we have stained a new batch of cortical brain samples (Brodmann area 6) from ALS (n=4), Alzheimer's disease (n=4), Multiple Sclerosis (n=4), and non-neurological controls (n=4) for ASRGL1 expression with an automated staining device (Leica Bond-MAX automated slide Stainer), to avoid differences due to manual staining. Ten regions per slide were randomly selected. Quantification of ASRGL1-positive cells per region was performed automatically using Gen5 Image Prime software (BioTek, Winooski, VT). The percentage of ASRGL1 positive cells was lower in ALS individuals compared to the other pathologies and to non-neurological controls (ANOVA p-value=0.0004) (Figure 1b and 1e). Besides,

we stained contiguous slides of the same ALS samples for ASRGL1 and TDP-43 to measure the percentage of cells with cytoplasmic TDP-43. We correlated it with the number of ASRGL1+ cells. We found a negative correlation between the number of ASRGL1+ cells and the presence of cytoplasmic TDP-43 (Pearson $r=-0.84$; p -value=0.008) (Figure 1d). These results have been included in the manuscript (page, 7, line 15- page 8, line 7).

2. Also, how is expression in other unaffected regions, such as the cerebellum?

- Response: We stained brain samples from Brodmann area 6 (pre-motor cortex) and Brodmann area 17 (visual cortex) from the same ALS individuals. While the percentage of ASRGL1+ cells was lower in BA17, the differences were not statistically significant (Figure 1c and 1e; manuscript page 7, lines 20- page 8, line 7).

3. The staining of patient and control brains in figure 1c is not fully convincing. ALS cases appear to have a generally lower DAB staining.

- Response: As stated in response to question 1, we stained new brain samples with an automated staining device to avoid differences due to manual staining. The new samples have a similar DAB background and ALS brains again show a lower percentage of ASRGL1+ cells (Figure 1b and 1e).

4. Figure 1h-l. It appears that nearly 100% of neurons and glia have TDP-43 mislocalization. This is not in line with previously reported, and TDP-43 pathology needs to usually be carefully searched to be found. This contradiction with literature and pathology practice needs to be resolved.

- Response: Yes, we agree that the percentage of cells showing cytoplasmic TDP-43 was striking, because for TDP-43 staining by immune fluorescence we selected samples with the lowest ASRGL1 levels. Now we have stained chromogenically, randomly selected ALS samples for both ASRGL1 and TDP-43 with an automated device. The percentage of cells showing cytoplasmic TDP-43 is now lower and we found a negative correlation between the levels of ASRGL1 and the presence of cytoplasmic TDP-43 (Pearson $r=-0.84$; p -value= 0.008) (Figure 1d and supplementary figure 4). The previous samples stained by immunofluorescence have been included in another figure (Figure 2), clearly stating that these samples were pre-selected for low levels of ASRGL1.

5. In Figure 2i, it appears that levels of interaction between TDP-43 and ASRGL1 in ALS are similar to background. Have technical controls been always performed?

- Response: Yes, technical controls were always performed. The low level of interaction detected is probably due to the low level of ASRGL1 in those samples.

6. It would be great to see the data from the mutant iPSC. These have been very rarely reported to have TDP-43 pathology, which should instead occur if ASRGL1 is reduced.

- Response: Unfortunately, we only had lysates from those iPSC-derived motor neurons. We could not get more of those cells to differentiate them into neurons and perform the TDP-43 staining. We will certainly make this referee's suggestion a priority for future studies.

7. Does AZT (and also re-expression of ASRGL1) impact on TDP-43 mislocalization?

- Response: The experiment with AZT was done in neural stem cells, not in neurons, because we wanted to decrease the expression of HERV-K (HML-2), which is considerably high in stem cells but very low in healthy neurons. We have performed an additional experiment to test whether the re-expression of ASRGL1 could rescue the viability of neurons and the TDP-43 pathology. We transfected neuronal cultures with a TDP-43 encoding plasmid and with shRNAs against ASRGL1 and co-transfected them with increasing concentrations of an ASRGL1-encoding plasmid or with pcDNA as a control. Then, by immunofluorescent staining we analyzed the number of cells showing cytoplasmic TDP-43 by confocal microscopy. Around 15% of neurons knocked-down for ASRGL1 showed cytoplasmic TDP-43, while cells transfected with increasing concentrations of the ASRGL1 plasmid, showed significantly lower percentage of cells with the cytoplasmic aggregates. Rescue of cell viability could also be seen in the neurons transfected with the two highest concentrations of ASRGL1 plasmid compared to the control cells. These results have been included in the main text (page 17, lines 1-10) and in Figure 5.

8. Correlation between ASRGL1 and HML-2 does not appear to be meaningful (Figure 5i).

- Response: We agree that the correlation is not strong, but the graph shows it is a significant correlation ($p=0.01$) and has a correlation coefficient $r=-0.40$, which is usually considered a moderate degree of association (Schober et al. Correlation Coefficients: Appropriate Use and Interpretation. Anesthesia & Analgesia 126(5):p 1763-1768). Considering the limited number of samples, we think it can be relevant and worth exploring it in larger cohorts in the future. We have updated the text to clearly state that there was a weak to moderate degree of the correlation (page 20, lines 3-5).

Reviewer #2 (Remarks to the Author):

In this manuscript the authors investigate whether the loss of asparaginase-like 1 protein (ASRGL1) contribute to the development of ALS by directly affecting TDP-43 protein misfolding, impaired degradation and consequently cytoplasmic aggregation. Furthermore, HERV-K (HML-2) overexpression (also previously reported in ALS by the same group) is suggested to induce

such ASRGL1 silencing. The manuscript proposes a model that encompasses and links together diverse findings/mechanisms to reach this conclusion: 1) ASRGL1 is decreased in the cortex of ALS patients which is accompanied by cytosolic aggregation of TDP-43; 2) the loss of ASRGL1 is responsible for increased isoaspartate formation of TDP-43 and, when present, its interaction with TDP-43 in the perinuclear region promotes protein degradation and protein aggregation in the cytoplasm is prevented; 3) silencing of ASRGL1 with shRNA or by activating HML-2 leads to neuron death in mice and neuronal cell cultures.

The manuscript is well written and easy to understand and the hypothesis put forward by the authors are interesting and novel. However, the presented data is overall not fully convincing and there are several control experiments are missing. Overall, the manuscript appears to preliminary in its current form to warrant publication.

Specific comments

1. The RNA-seq analysis in Fig 1 is convincingly showing that ASRGL1 is downregulated in ALS tissue. However, single-cell analysis (using for example available snRNA-seq data from ALS tissue) would be a very valuable addition to clarify in which cells ASRGL1 is expressed in control individual and in what cell-types ASRGL1 expression is reduced in ALS.

- Response: In collaboration with researchers at the Massachusetts Institute of Technology (MIT) we have performed analysis of single cell sequencing data of a cohort of 73 individuals (sALS; n=17, C9orf72-associated ALS; n=16, sporadic FTLD; n=13, C9orf72-associated FTLD; n=11 and non-neurological controls; n=16). The results show that ASRGL1 expression is reduced in several neuronal types in layer 5 (the layer most affected in ALS) and 4, and specially in astrocytes in sporadic ALS and C9orf ALS, although sub-threshold. TARDBP is significantly elevated in the C9orf ALS group in excitatory neurons from layers L2-L3 and L3-L5, L4-L6 and L5-L6, in microglia, oligodendrocytes precursor cells (OPCs), and in the inhibitory neurons In5HT3aR VIP- and PV Basket. TARDBP is elevated in the sporadic ALS group in several subtypes of neurons but without reaching statistical significance. It is important to note that single-cell data is poor at capturing relatively low-expressed genes, and both ASRGL1 and TDP43 are expressed at low levels, so it is not surprising that the changes did not reach statistical significance. The MIT protocol is also very stringent in how they classify differentially expressed genes (DEGs), as this type of data is also very noisy, so they are more likely to have false negatives than false positives. As for HERV-K (HML-2), the MIT team has encountered a similar issue. Most transposable elements (TEs) are not captured by this assay, and those which are, are not sufficiently abundant to analyze. The assay only recovers ~90 bases from the 3' end, which is not sufficient to accurately identify many of these transcripts. This analysis has been described in the main text (page 6, line 4-page7, line 10) and in Supplementary figures 2 and 3.

2. The IHC pictures are not very convincing. For example, how come there seems to be a

complete absence of staining in ALS samples, while the western blot in (Fig 1 d, e) show a difference but far from being that drastic decrease.

- Response: IHC images in the previously submitted manuscript were selected to show the biggest differences and western blot is far more sensitive for detection because a bigger piece of tissue is analyzed. To confirm the previous results, we have stained other brain samples to include additional control groups and brain areas. We have stained cortical brain samples (Brodmann area 6) from ALS, Alzheimer's disease, Multiple Sclerosis, and non-neurological controls for ASRGL1+ with an automatic staining device (Leica Bond-MAX automated slide stainer; Leica Biosystems), to avoid differences due to manual staining. Quantification of ASRGL1-positive cells per region was also performed automatically using Gen5 Image Prime software (BioTek, Winooski, VT). With this new methodology the percentage of ASRGL1 positive cells is still lower in ALS individuals compared to the other pathologies and to non-neurological controls (ANOVA p-value=0.005). We have also compared ASRGL1 staining in BA6 and BA17 areas from the same ALS individuals. The manuscript has been updated to include these results in the main text (page 7, line 15- page 8, line 7) and in Figure 1.

3. The use of iPSC-derived motor neurons is an excellent idea. However, a more thorough characterization of this iPSC derived motor neurons cultures would be helpful for the interpretation of the data. Are they lower motor neurons (referenced protocol says 30%), what are the other cell types present?

- Response: The cells were seeded as aggregated spheres to get better motor neuronal differentiation. 95% of the differentiated neurons (with long neurites, grew out from the neurospheres) were ChAT positive, indicating a lower motor identity. These cells were also ISL-1 positive, to a lesser degree. We could not identify the other cells (<5%), but based on their morphology, they appear to be glial cells, though at this stage they were GFAP negative. It is difficult to clarify the cell types in the center of the aggregated neurospheres precisely as they were compact, but most were undifferentiated neural stem/progenitor cells with positive nestin staining. Characterization of the motor neurons has been included in Material and Methods (page 47, line 19- page 48, line 2) and in Supplementary Figure We have added this information in page 8, lines 21-23 and in Supplementary figure 6.

4. The authors rely heavily on quantification of cell viability. However, in order to estimate cell viability there are several methods available which all have their pros and cons. Since this experimental output is essential in the manuscript the measure by fluorescence microscopy is not convincing. In addition, it is not clear to the reader what exact measurement is acquired with fluorescence microscopy. Please clarify, Cell count? Neurite length? Could the authors think of an additional approach to address this? For example, Flow cytometry or simple IP staining...? It is also confusing that Fig. 3s and t have a different nomenclature if the

measurements for viability are the same as for the other panels in the manuscript addressing cell viability.

- Response: Panels have been re-labeled to clarify what is measured in each case. Besides, we have repeated the experiments to use a different method to measure cell viability. In this case, we have used Alamarblue (Thermofisher), which is a ready-to-use resazurin-based solution that functions as a cell health indicator by using the reducing power of living cells to quantitatively measure viability. Resazurin, the active ingredient of AlamarBlue reagent, is a non-toxic, cell-permeable compound that is blue in color and virtually non-fluorescent. Upon entering living cells, resazurin is reduced to resorufin, a compound that is red in color and highly fluorescent. After staining the cells with this compound, viability was measured with a fluor spectrometer (Molecular Devices). The new results on cell viability are shown in Figure 5, Supplementary figure 9 and described in the text (page 13, lines 10-16).

5. The results from the shRNA-experiments are intriguing. However, a lot of detail is missing. How were the shRNA-constructs validated. Rescue-experiments to verify a lack of off-target effects are also missing.

- Response: Silencing of ASRGL1 by the shRNAs was validated measuring a decrease in RNA expression by PCR and in the protein expression by western-blotting. A figure showing these results has been added (Supplementary figure 10) and in the text (page 13, lines 15-16). Besides, we have also performed rescue experiments as suggested by the reviewer. We transfected neuronal cultures with a TDP-43 encoding plasmid and with shRNAs against ASRGL1 which resulted in mislocalization of TDP-43 and neurotoxicity. Both parameters recovered when we co-transfected the cells with increasing concentrations of a ASRGL1-encoding plasmid in a dose-responsive manner. These results have been included in a new figure (Figure 5a and 5b) and described in the text (page 17, lines 2-10).

6. The authors mention two ways of isoaspartate formation but only address asparagine deamidation experimentally, why?

- Response: In-silico analysis by NGOME-Lite software shows that only asparagine deamidation is relevant for isoaspartate formation in TDP-43. NGOME could not detect any rapid isomerization of aspartate residues to isoaspartate. This is because (1) Asp-Gly dipeptides isomerize slower than most Asn-Xaa dipeptides deamidate and (2) Asp-Gly dipeptides in TDP-43 are found in folded regions protected from isomerization reactions. These results have been added to the manuscript (page 10, lines 20-23).

7. In fig. 2i low values for PLA are shown, how does the authors know that these values are related to the proximity between the protein and not just a reflection of the lower ASRGL1

expression, something already addressed in figure 1. If the authors want to point towards a proximity the samples are supposedly normalized, however this is not clear.

- Response: We apologize for the lack of clarity. The decrease in PLA signal shown in figure 2i would be another way to validate the lower levels of ASRGL1 found in ALS brains. We have updated the text for better clarity (page 12, lines 14-16).

8. Why is there not a control for ASGL1+IgG in Fig.2 f and i?

- Response: We considered it was unnecessary to analyze the interaction because the sum of the backgrounds of the TDP-43 and ASRGL1 alone are far from what it is seen when both are used together in the control brains, which indicates the specificity of the interaction signal.

9. In previous research the same authors report that “expression of HERV-K was regulated by TDP-43” Li et al. 2015. This aspect is not addressed in this manuscript; however this need to be properly discussed in relation to the suggested activation of HERV-K by ASRGL1. Are both factors involved in the HERV-K expression, etc...?

- Response: Thank you for the suggestion. A paragraph has been included in the Discussion section to discuss the possible complex relationship that might exist between TDP-43 and ASRGL1 (page 25, lines 14-20).

10. It is important that the authors provide more information about other factors affected by the HML-2 activation in Fig. 5, essentially but not exclusively proteins related to the HERV-K family and TDP-43.

- Response: A paragraph has been included in the Discussion (page 24, lines 14-22).

11. It is hard to follow what models (cell type, human brain tissue etc.) are used in which panels in most figures. Please clarify.

- Response: We have updated the Results section and the Figure legends to clearly state the models that were used.

12. The discussion is hard to follow and disperse and would need editing.

- Response: The discussion has been edited. To improve the flow and understanding, we have also added some subheadings.

13. On line 246 the authors claim that “TDP-43 levels decreased in the neurons treated with scrambled shRNA” which is an over interpretation of the data.

- Response: The text has been updated to state that TDP-43 levels did not change significantly in the neurons treated with scrambled shRNA (page 16, line 10).

14. In Fig. 3 there is a difference in the protein levels in the TDP-43 western blots comparing 3e,g with 3o at 0h. This needs to be explained since a variation in total TDP-43 expression would affect the phosphorylated TDP-43 results.

- Response: In the experiments shown in figure 3e and 3g, the neuronal cultures were transfected with a plasmid encoding TDP-43 (besides the intrabody in figure 3e and the ASRGL1 shRNAs) while in the experiment shown in figure 3o, the neuronal cultures were not transfected with TDP-43, therefore the levels shown in the blots are lower than in 3e and g and only due to endogenous expression. The text has been updated to clearly state this distinction (page 16, lines 8-9 and page 15, lines 5-6). Besides, it is important to note that for the measurement of misfolded TDP-43 and TDP-43 fragments, the blots had to be overexposed, due to the low levels of those pathological proteins. This has also stated clearly in the text (page 15, lines 5-7).

15. Fig 3: In several panels it is vague what % corresponds to.

- Response: Panels have been re-labeled to clarify this.

16. The n=values are hard to understand.

- Response: Text has been updated to clarify whether n corresponds to number of samples or number of replicates in each experiment.

17. In Fig. 2f the dots are red for ALS. Should they be black?

- Response: We found the mistake in the color of the dots in figure 1f. Figure 1 has been updated and the error fixed.

18. On line 191 the figure is mis-referenced. Should it be Fig 2g?

- Response: Yes, thank you. The error has been fixed (page 12, line 12).

19. The reference numbers should be in superscript on line: 215, 225, 241, 279, 302, 310, 318, 362, 420, 447.

- Response: Yes, thank you. The typos have been fixed.

20. Reference 30 needs editing.

- Response: Corrected. Thank you.

21. On line 1213 the 3 is missing in TDP-43.

- Response: Corrected. Thank you.

Reviewer #3 (Remarks to the Author):

In this study, Garcia-Montojo and colleagues identify loss of asparaginase-like 1 (ASRGL1) as a potential explanation for TDP43 pathology and neurodegeneration in ALS. The same group previously demonstrated activation of endogenous retroviruses such as HML2 in ALS samples. As shown in this manuscript, because the gene encoding ASRGL1 contains an HML2 copy, reactivation of virus leads to ASRGL1 downregulation at the RNA and protein levels. The authors also show that TDP43 is an ASRGL1 substrate. In the absence of this enzyme TDP43 becomes misfiled, accumulates in the cytoplasm and is phosphorylated, similar to what is found in ALS patients. Loss of ASRGL1 is also sufficient to reproduce neurodegeneration in cultured cells, and ASRGL1 overexpression protects against toxicity arising from TDP43 accumulation.

The authors provide a compelling argument for the connection between ASRGL1, TDP43 and neurodegeneration in ALS though a combination of in silico, in vitro, in vivo, post mortem, and cellular models. In general this is an exciting finding that could provide an explanation for TDP43 pathology in the setting of retrovirus activation. Even so, there are some concerns that could strengthen the findings if addressed.

Major concerns:

1. More information is required surrounding the detection of TDP43 asparagine deamination in ALS brains by mass spectroscopy. Is there a relationship between deamination and sex, since the proportion of males is higher in controls vs. ALS samples? How commonly were specific residues deamidated in controls vs. ALS patients? Are the effects region specific? How does this compare to other proteins, i.e. is the effect specific to TDP43?

- **Response:** The sample sizes are too small to determine any sex or regional specific differences in deamination. But this is an excellent suggestion for future studies.

2. Fig. 2i -- this is interesting and implies that TDP43 interacts with ASRGL1 in human brain. It would be ideal if this interaction could be confirmed by a separate method (i.e. Co-IP), although this reviewer understands that such assays can be problematic.

- **Response:** Thank you for this suggestion. We have tried co-immunoprecipitation in brain samples and, although we got positive results in several samples, we think the results are not consistent enough for publication. The reason might be that, since ASRGL1 is a protease, the interaction of both proteins might be too transient to detect it consistently by this method (once the proteolysis occurs, the two proteins might not interact anymore). We have added a sentence to the results section (page 12, lines 20-22). You can see a blot of a Co-IP below:

3. Additionally, the results could be confounded by the reduction in ASRGL1 protein shown in Fig. 1. Is the PLA signal region dependent, meaning higher in affected regions of the CNS vs. unaffected?

- **Response:** Yes, thank you for pointing this out. We have only performed PLA assay in the cortex. See response above to comment 7 by the 2nd reviewer. The lower PLA signal detected in this experiment is likely due to the lower levels of ASRGL1 in ALS samples. We have updated the text for better understanding (page 12, lines 14-16).

4. Fig. 3e-j -- the authors are encouraged to show efficiency of ASRGL1 knockdown in transfected cultures. It is surprising that they were able to see such robust results by WB after transient transfection, which is not always so efficient in these cultures.

- **Response:** We have added the validation of ASRGL1 knockdown by the shRNAs as measured by PCR and western blotting in Supplementary figure 10.

5. Fig. 3l -- The authors are encouraged to measure signs of TDP43 dysfunction as indicated by cryptic exon inclusion in STMN2 or UNC13A, in human neurons upon ASRGL1 knockdown.

- Response: Thank you very much for this suggestion. In neuronal cultures transfected with the ASRGL1 shRNAs or with scrambled shRNAs, we have measured the cryptic exon of UNC13A by PCR and UNC13A protein by western blotting. The results show that ASRGL1 knockdown causes an increase in the levels of the cryptic exon and a consequent decrease in the protein levels, indicating a dysfunction in its splicing, which is driven by TDP-43. This experiment is shown in Figure 4m, n, o and described in the text (page 15, lines 12-20).

6. Fig. 3o,p -- the results showing dramatic accumulation of TDP43 after only 6h are surprising. This is significantly more than what is seen with impairment of the proteasome by MG132 (which appears to have little or no effect compared to Sc shRNA). How do the authors account for this?

- Response: Although the effect of ASRGL1 knockdown on the accumulation of TDP-43 is indeed highly significant compared to the cells transfected with scrambled shRNA, it is difficult to compare it with the effect of MG132. MG132 was applied 3 hours prior to adding the cycloheximide, so an accumulation of TDP-43 is already seen at 0h of the cycloheximide experiment in the MG132 treated cells. An increase in the loading control vinculin could also be seen at this time point (Figure 4r). Due to this, change in TDP-43 in these cells could not be accurately calculated. This information has been added to the manuscript (page 16, lines 12-17).

7. Fig. 3s,t -- how do the authors explain toxicity from TDP43 peptides? This is an unconventional way to introduce TDP43 toxicity, and its relevance is unclear.

- C-terminal fragments of TDP-43 have been shown to be toxic to the neurons by transfection and in stably CTF-25 expressing motor neurons¹. They have been shown to accumulate in the cytoplasm of neurons in ALS. We hypothesize that one of the reasons for their accumulation in ALS might be the loss of ASRGL1. ASRGL1 degrades isoaspartyl peptides from the peptide chains where they are naturally produced by asparagine deamination. Loss of ASRGL1 would lead to an accumulation of those detrimental isoaspartyl peptides in TDP-43, which would impair their degradation. In this experiment, we tried to mimic the neurotoxicity caused by those isoaspartyl peptides when ASRGL1 is knocked down, and determine if recovery in ASRGL1 expression can rescue the toxicity. Isoaspartate formation is a naturally occurring post-translational modification, so we cannot mimic it by transfection with a plasmid. The best way to test our hypothesis was to transfect with peptides harboring the posttranslational modification. This has been clarified in the manuscript (page 17, lines 11-22).

8. Fig. 4e -- TDP43 mislocalization in ASRGL1 shRNA-treated animals is likewise impressive. Additional evidence of TDP43 dysfunction as judged by missplicing could strengthen this result (see for example PMID 29764981).

- Response: Thank you again for this suggestion. As you suggested in comment #5, we have performed this experiment in vitro in neuronal cultures by knocking-down ASRGL1 with shRNAs and the results show a decrease in UNC13A protein expression and an increase in the detection of the cryptic exon, which would indicate a dysfunction in the splicing of this gene by TDP-43. These results are very encouraging, and we would like to study this phenomenon in the animal model. Unfortunately, we did not have frozen tissue from the mice treated with the AAV viral particles carrying the ASRGL1 shRNAs to conduct it now. We will perform this in future studies.

9. Fig. 5k, l -- these plots show that in some patients with ALS (but not all or even a majority) HML2 RNA levels are higher than controls. It is difficult to conclude that HML2 increase is a common driver of neurodegeneration in ALS with this degree of variability.

- Response: We agree. HML-2 overexpression might contribute to ALS pathology in a subset of patients. In this and in previous studies we have detected it in approximately 30% of the patients. Loss of ASRGL1 seems to be a more common feature than the increase in HML-2, which supports the idea that loss of ASRGL1 might have different triggers in different ALS individuals, such as mutations in ASRGL1 gene or in its transcription factors. This has been included in the discussion (page 24, lines 18-22).

10. The other challenging aspect of this plot is that if only ALS patients are considered (pink dots), there is no real relationship between HML2 RNA and ASRGL1 protein levels. This is inconsistent with the authors' hypothesis. Probably because the small sample size.

- Response: As mentioned above, HML-2 overexpression is mostly restricted to a subset of ALS patients (although it can be found in some controls). This makes a correlation difficult because there is not enough dispersion in one of the variables due to the limited number of samples. Despite this, a trend for an association between the two variables could still be seen in ALS patients (Spearman's $r = -0.36$; $p = 0.12$) and not in controls (Spearman's $r = 0.06$; $p = 0.80$) when we separate both groups (except for one

control that has also high HML-2 and low ASRGL1).

Minor:

1. TDP43 phosphorylation likely has minimal effects on TDP43 trafficking or function, and if anything counteract aggregation (see PMID 35112738).
 - Response: Thank you for this reference. We agree that the role of TDP-43 hyperphosphorylation in ALS remains unclear, with some authors suggesting that it might be a protective cellular response to counteract TDP-43 aggregation. Despite this role, it is one of the hallmarks of ALS. We have updated the manuscript to clarify this, and we have included this reference (page 14, line 23- page 15, line 3).
2. Fig. 1g and 1h -- please describe the origin of cells (i.e. sALS) within the text or figure legend.
 - Response: Descriptions of the origin of cells have been added to the figure legends and the main text.
3. Fig. 3a-d -- the efficacy of ASRGL1 knockdown in each case and combination should be confirmed.
 - Response: Graphs with the effect of individual shRNAs on neurotoxicity have been added in Supplementary figures 8 and 9. Validation of the ASRGL1 knockdown by PCR and western blotting with the combination of shRNAs used in the proteinopathy experiments has been included in Supplementary figure 10. To determine if the neurotoxicity was caused by off-target effects of the shRNAs, we have also performed rescue experiments which show recovery of neuronal viability and TDP-43 pathology after restoration of ASRGL1 expression in a dose-responsive manner (Figure 5a and 5b).
4. Fig. 3l -- knockdown of ASRGL1 does not appear to be very efficient in the indicated cells with TDP43 mislocalization.
 - Response: As presented in Supplementary Figure 10, the combination of shRNAs used in the experiments does not completely knockout ASRGL1 expression. We selected the mentioned image intentionally to show cells with and without expression of ASRGL1 to see the difference in TDP-43 localization (Fig. 4l).

5. Sup. Fig. 3 -- why do neurons have cytoplasmic TDP43? Have they been stressed in some way, or is this a dead/dying cell?
- Response: Supplementary figure 3 (updated to figure 2f now) shows an amplified image of brain cells in a cortex sample of an ALS patient, to show how loss of ASRGL1 is associated with cytoplasmic TDP-43 in the brain.
6. Fig. 5k, l -- the y-axes on these plots are not intuitive. For 5k, what is the 'HML2 RNA fold change' measured in relation to? And for 5i, the legend and text describe ASRGL1 protein levels, but the plot shows 'ASRGL1 ratio'.
- Response: "Fold change" in all graphs presenting RNA expression is calculated in relation to the average of the controls, according to the Delta Delta Ct method (Livak method²). In the case of figure 5k, the fold change has been calculated with the average of the non-neurological control samples. In all graphs presenting protein expression, the calculation is done by a ratio between the intensity of the band of the target protein and the intensity of the band of the loading control protein, as measured by western blotting. We have added this information in the methods section (page 57, lines 6-9 and page 63, lines 3-5) and updated the labelling of the panels and the figure legends to homogenize the terms.
7. In the discussion, the authors stated that 'ASRGL1 rescues the neurotoxicity caused by TDP-43 C-terminal fragments' (line 417-418). This is not entirely accurate, since ASRGL1 expression rescued toxicity from a small peptide that corresponds to part of the TDP43 C-terminus. It remains unclear whether ASRGL1 can prevent toxicity from full length TDP-43 or CTFs.
- Response: A clarification has been included in the text (page 17, lines 11-14).

References

- 1. Liu, Y., *et al.* A new cellular model of pathological TDP-43: The neurotoxicity of stably expressed CTF25 of TDP-43 depends on the proteasome. *Neuroscience* **281**, 88-98 (2014).
- 2. Livak, K.J. & Schmittgen, T.D. Analysis of relative gene expression data using real-time quantitative PCR and the 2(-Delta Delta C(T)) Method. *Methods* **25**, 402-408 (2001).

Reviewer #1 (Remarks to the Author):

The authors have addressed multiple points by performing new single cell analysis and revisiting their staining protocols and experiments. Unfortunately they could not provide work in mutant iPSCs which would have been useful, but this appears to be limited by no availability of material. Generally the manuscript is improved.

Reviewer #2 (Remarks to the Author):

The authors have addressed most referee comments, and the manuscript is greatly improved. There is now clear evidence of the involvement of ASRGL1 and TDP-43 proteinopathy and their relation in the development of ALS. The additions and refinements made to Figures 1-6 are convincing, and the additional work has significantly strengthened the overall presentation of these findings. However, the last part of the manuscript (the data presented in figure 7) has not been much improved. The link between the retroviral element HERV-K and ASRGL1 remains undefined. The data presented for HERV-K currently lacks the clarity and precision needed to support a firm connection. To enhance the robustness of this crucial aspect, substantial additional work on this part to match the high standard set by the rest of the manuscript is needed.

An alternative, which may be to prefer, is to remove the data in figure 7 from the manuscript. Since the data in this section does not significantly contribute to the central theme and fails to provide a conclusive link, the option of removal may be worth considering. Such a decision would contribute to the overall clarity and focus of the manuscript. Although the exclusion of this section would significantly impact the central focus of the manuscript, as indicated by the title, we believe that this would be beneficial to maintain the overall coherence and strength of the study.

Minor comment:

The control data presented from the western blots in several figures are all presented without standard deviation, what we can see. The use of one-way ANOVA, which assumes homogeneity of variances thus, would not be well suited for this data. To the reviewers' understanding and as seen in some supplementary data you could potentially calculate relative normalized ratio before one-way ANOVA or, use e.g. Welch test (Welch ANOVA). If this is already true, it should be explicitly mentioned in the text.

Reviewer #3 (Remarks to the Author):

The authors have addressed most of my concerns, and added quite a bit of new data to the manuscript strengthening their original conclusions.

REVIEWER COMMENTS

We are delighted that reviewers 1 and 3 felt that we had addressed all their comments and that no further changes were needed. Reviewer 2 also felt that the manuscript was greatly improved but had two additional comments which we have addressed below.

Reviewer #2 (Remarks to the Author):

The authors have addressed most referee comments, and the manuscript is greatly improved. There is now clear evidence of the involvement of ASRGL1 and TDP-43 proteinopathy and their relation in the development of ALS. The additions and refinements made to Figures 1-6 are convincing, and the additional work has significantly strengthened the overall presentation of these findings.

However, the last part of the manuscript (the data presented in figure 7) has not been much improved. The link between the retroviral element HERV-K and ASRGL1 remains undefined. The data presented for HERV-K currently lacks the clarity and precision needed to support a firm connection. To enhance the robustness of this crucial aspect, substantial additional work on this part to match the high standard set by the rest of the manuscript is needed.

An alternative, which may be to prefer, is to remove the data in figure 7 from the manuscript. Since the data in this section does not significantly contribute to the central theme and fails to provide a conclusive link, the option of removal may be worth considering. Such a decision would contribute to the overall clarity and focus of the manuscript. Although the exclusion of this section would significantly impact the central focus of the manuscript, as indicated by the title, we believe that this would be beneficial to maintain the overall coherence and strength of the study.

- Response: Thank you very much for your comments. We really appreciate your insight. We agree that there is still substantial work needed to fully elucidate the relationship between ASRGL1 and HML-2, since the mechanism of downregulation of ASRGL1 by HML-2 is still unclear. Novel contributions of this study result from the observation of the presence of a copy of the HML-2 gene in an antisense orientation in the intron of ASRGL1. We observed that there was an inverse relationship between the expression of HML-2 and ASRGL1 in ALS brains; overexpression of HML-2 resulted in down regulation of ASRGL1 in neurons in a dose-responsive manner which was also confirmed by inducing the expression of endogenous levels of HML-2. Conversely over expression of ASRGL1 rescued HML-2 induced neurotoxicity. Furthermore, inhibition of reverse transcriptase resulted in an increase of ASRGL1 in a dose-responsive manner. Together, these observations clearly establish a functional relationship between the two genes. Perhaps we did not explain the findings and their relevance well. We have now made revisions to the title, abstract, results and discussion accordingly. We have emphasized the functional association and have been careful not to imply causation. At the same time, we have also clearly stated that much work needs to be done to further elucidate the interactions between the two genes and suggested possible mechanisms that could be explored. We hope you will agree that inclusion of these observations will advance the field and stimulate further research.

Minor comment:

The control data presented from the western blots in several figures are all presented without standard deviation, what we can see. The use of one-way ANOVA, which assumes homogeneity of variances thus, would not be well suited for this data. To the reviewers' understanding and as seen in some

supplementary data you could potentially calculate relative normalized ratio before one-way ANOVA or, use e.g. Welch test (Welch ANOVA). If this is already true, it should be explicitly mentioned in the text.

- Response: Thank you very much for your comment. In the previous version of the manuscript, ANOVA was chosen after applying tests to check the homogeneity of variances (Barlett's test) and only if those did not show statistical significance. However, following your suggestion, we have redone all the statistical analyses that involved the comparison between 3 or more groups with evident differences in variances using the Brown-Forsythe ANOVA test, which is similar to Welch's but more appropriate when the sample is not gaussian¹ (https://www.graphpad.com/guides/prism/latest/statistics/interpreting_welch_browne-forsythe_tests.htm). For the multiple comparisons, we have applied Bonferroni corrections (<https://universeofdatascience.com/heteroscedastic-anova-tests-in-r/>), which is the most conservative of all the post-hoc tests, to be sure that the results were indeed significant. For the comparison between two groups of different variances, we have applied the non-parametric test Mann-Whitney's. The legends of the figures have been updated with the statistical tests used, as well as the figures and the Material and Methods section (page 63, lines 7-10).

Bibliography

- 1. SA Glantz, B.S., TB Neilands. *Primer of Regression & Analysis of Variance*, (2016).

Reviewer #2 (Remarks to the Author):

I have no further comments. However, the authors did respond satisfactory to any of my comments from the last round of revisions.

Response to the reviewers' comments

We are delighted to learn that the manuscript has been accepted for publication. We would like to thank the referees for their insight and suggestions. They have undoubtedly helped to make our manuscript much more compelling.